# Hydrodynamic and Biochemical Impacts on the Development of Hypoxia in the Louisiana–Texas Shelf Part II: Statistical Modeling and Hypoxia Prediction

Yanda Ou[1] , Bin Li[2],  Z. George Xue[1,3,4]

[1]Department of Oceanography and Coastal Sciences, Louisiana State University, Baton Rouge, LA, 70803, USA.

[2]Department of Experimental Statistics, Louisiana State University, Baton Rouge, LA, 70803, USA

[3]Center for Computation and Technology, Louisiana State University, Baton Rouge, LA, 70803, USA.

[4]Coastal Studies Institute, Louisiana State University, Baton Rouge, LA, 70803, USA

*Correspondence to*: Z. George Xue (zxue@lsu.edu)

**Abstract.** This study presents a novel ensemble regression model for hypoxic area (HA) forecast in the Louisiana–Texas (LaTex) Shelf. The ensemble model combines a zero-inflated Poisson generalized linear model (GLM) and a quasi-Poisson generalized additive model (GAM) and considers predictors with hydrodynamic and biochemical features. Both models were trained and calibrated using the daily hindcast (2007–2020) by a three-dimensional coupled hydrodynamic–biogeochemical model embedded in the Reginal Ocean Modeling System (ROMS). Compared to the ROMS hindcasts, the ensemble model yields a low root-mean-squared error (RMSE) (3,256 $km^2$), a high $R^2$ (0.7721), and low mean absolute percentage biases for overall (29 %) and peak HA prediction (25 %). When compared to the Shelf-wide cruise observations from 2012 to 2020, our ensemble model provides a more accurate summer HA forecast than any existing forecast models with a high $R^2$ (0.9200), a low RMSE (2,005 $km^2$), a low scatter index (15 %), and low mean absolute percentage biases for overall (18 %), fair-weather summers (15 %), and windy summers (18 %) predictions. To test its robustness, the model is further applied to a global forecast model and produces HA prediction from 2012 to 2020 with the adjusted predictors from the HYbrid Coordinate Ocean Model (HYCOM). In addition, model sensitivity tests suggest an aggressive riverine nutrient reduction strategy (92 %) is needed to achieve the HA reduction goal of 5,000 $km^2$.

## 1 Introduction

The Louisiana–Texas (LaTex) Shelf has become a center of hypoxia (bottom dissolved oxygen, DO<2 mg $L^{-1}$) study since the 1980s (e.g., Rabalais et al., 2002; Rabalais et al., 2007a; Justić and Wang, 2014). Regular mid-summer Shelf-wide cruises documented that the area and volume of hypoxic bottom water could reach up to 23,000 $km^2$ and 140 $km^3$, respectively (Rabalais and Turner, 2019; Rabalais and Baustian, 2020). The aquatic environments, fisheries, and coastal economies are under threat of recurring hypoxia in summer (Chesney and Baltz, 2001; Craig and Bosman, 2013; De Mutsert et al., 2016; LaBone et al., 2020; Rabalais and Turner, 2019; Rabotyagov et al., 2014; Smith et al., 2014). For example, habitats of some fish species (e.g., croaker and brown shrimp) shift to offshore hypoxic edges (Craig and Crowder, 2005; Craig, 2012) during

summer hypoxia events, which may impact organism energy budgets and trophic interactions (Craig and Crowder, 2005; Hazen et al., 2009). The horizontal displacement of brown shrimp habitats in summer can also lead to changes in the distribution of Gulf shrimp fleets (Purcell et al., 2017). Although an Action Plan has been launched by the Mississippi River/Gulf of Mexico Hypoxia Task Force to control the size of the mid-summer hypoxic zone below 5,000 km$^2$ in a 5-year running average since 2001 (Mississippi River/Gulf of Mexico Watershed Nutrient Task Force, 2001; 2008), hypoxic areal extents experience no significant decreases in recent decades (Del Giudice et al., 2020). An accurate prediction of the hypoxic area is urgently needed for coastal managers and the fishery industry.

Water column stratification and sediment oxygen consumption (SOC) are two main factors regulating the formation, evolution, and destruction of bottom hypoxia from mid-May through mid-September (Bianchi et al., 2010; Conley et al., 2009; Fennel et al., 2011, 2013, 2016; Feng et al., 2014; Hetland and DiMarco, 2008; Justić and Wang, 2014; Laurent et al., 2018; McCarthy et al., 2013; Murrell and Lehrter, 2011; Rabalais et al., 2007b; Wang and Justić, 2009; Yu et al., 2015). The stratification inhibits bottom water reoxygenation, while SOC, induced by water eutrophication associated with high anthropogenic nutrient supplies by rivers, can lead to anaerobic benthic environments. Nevertheless, existing hypoxic area (HA) prediction models rely most on contribution from the nutrient load rather than hydrodynamic features. Turner et al. (2006) built a multiple linear regression model for summer HA prediction using the annual and May nitrogen flux (nitrate+nitrite) of the Mississippi River as the predictors. The model provides a robust annual prediction when no strong wind is present but overestimates the HA in windy years. Obenour et al. (2015) modeled HA using the empirical relationship between HA and bottom DO concentration derived from a Bayesian biophysical model. Their model accounts for primary biophysical processes solved for steady-state conditions, water transport, May total nitrogen loads by rivers, and parameterized water reaeration. Katin et al. (2022) further adjusted the Bayesian model by taking into account river flows, riverine bioavailable nitrogen loadings, and wind velocity in both summer (June–September) and non-summer (November–May) months. Summer riverine inputs are projected using non-summer riverine variables, river basin precipitation, and river basin temperature, while summer wind velocity is resampled from historical records from 1985 to 2016. Therefore, the season prediction model is known as a pseudo-forecast model since predictors in future stages only include riverine inputs. This model explains 71 % and 41 %–48 % of the variability in hindcast (Del Giudice et al., 2020) and geostatistically estimated HA (Matli et al., 2018), respectively. An additional Bayesian model applied to summer bottom DO prediction accounts for May total nitrogen loads, distance from the Mississippi River mouth, and downstream velocity (Scavia et al., 2013). The summer HA is determined by hypoxic length (HA=57.8 hypoxic length) derived from summer bottom DO concentration. The model explains 69 % of the variability in observed HA by the mid-summer Shelf-wide cruises. Mechanistic prediction methods have also been applied by Laurent and Fennel (2019) to develop a weighted mean forecast that is calibrated using May nitrate loads and three-dimensional hindcast simulations over the period 1985–2018. Once calibrated, the model only requires May nitrate loads as an input to produce the seasonal forecast for a given year. The model can explain up to 76 % of the year-to-year variability of the HA observation. However, the model is not favorable for years with strong wind events during summer.

These above-mentioned models share some similar drawbacks. (1) The effects of water column stratification are considered
only implicitly by the associated wind speeds, water transport, and riverine nutrient loads (usually correlated to river
discharges), although stratification is documented as a crucial factor in regulating HA variability. (2) Forecast of the predictors
is usually limited, which restricts some of these seasonal models to pseudo ones. (3) Most models are only capable of capturing
interannual HA variability and are not reliable in summers when winds are strong. According to the hindcast results by our
three-dimensional coupled hydrodynamic–biogeochemical model described in the accompanying paper (Part I), strong wind
events bring considerable uncertainties to monthly and daily variabilities of HA. In this study we aim to provide a novel HA
prediction method that considers both stratification and biochemical effects. Our new model aims to produce daily HA
forecasts based on selected predictors' forecasts with a minimum computational cost. The rest of the paper is organized as
follows. Detailed descriptions of methods and data are given in section 2. The employments of generalized linear models
(GLMs), generalized additive models (GAMs), and an independent model application using a global forecast product (HYbrid
Coordinate Ocean Model, HYCOM; Bleck and Boudra, 1981; Bleck, 2002) are given in section 3. Comparisons against
existing forecast models, recommendations on nutrient reduction strategy, and model improvement outlook are given in section
78  4.

**2 Methods**
**2.1 Data preparation**
We adapted a three-dimensional coupled hydrodynamic–biogeochemical model embedded in the framework of the Regional
Ocean Modeling System (ROMS) on the platform of Coupled Ocean–Atmosphere–Wave–Sediment Transport Modelling
system (COAWST, Warner et al., 2010) to the GoM (Gulf–COAWST, for detailed descriptions, validations, and results of the
numerical model see Part I). Numerical hindcasts (hereafter denoted as ROMS hindcasts or ROMS simulations) are output
daily from 1 January 2007 to 26 August 2020 and spatially averaged over the LaTex Shelf extending from the west of
Mississippi River mouth to 95°W with water depths ranging from 6 to 50 m (color shaded region in Figure A1b).
**2.1.1 Hydrodynamic-related predictors**
Both water stratification and bottom biochemical processes modulate the variability of bottom DO concentration in the LaTex
Shelf. Potential energy anomaly (PEA, in J m$^{-3}$) is introduced as an estimate of water column stratification according to:

$$PEA = \frac{1}{H}\int_{-h}^{\eta}(\bar{\rho} - \rho)gzdz, \tag{1}$$

where $\rho$ is water density profile (estimated by water temperature and salinity profiles) over water column of depth $H = h + \eta$,
$h$ is the location of the bed, $\eta$ is water surface elevation, $g$ is the gravitational acceleration (9.8 m s$^{-2}$), $z$ is the vertical axis, $\bar{\rho}$
is the depth-integrated water density given by $\bar{\rho} = \frac{1}{H}\int_{-h}^{\eta}\rho dz$ (Simpson and Hunter, 1974; Simpson et al., 1978; Simpson,
1981; Simpson and Bowers, 1981). The PEA represents the amount of energy per volume required to homogenize the entire
water column (Simpson and Hunter, 1974). Thus, a greater PEA value represents a more stratified water column. As a river-
dominated area, water stratification in the LaTex Shelf is highly affected by freshwater-induced buoyancy from the Mississippi
and Atchafalaya Rivers. Sea surface salinity (SSS) is a good proxy for representing the distribution and variability of river
freshwater across the shelf. Indeed, the correlation of regionally averaged PEA and SSS is significantly high as -0.87 ($p<0.001$;
Figure 1a) which emphasizes the importance of freshwater-induced stratification. Therefore, we considered SSS as another
candidate predictor besides PEA.

Surface heating and wind mixing are two other factors that influence water stratification (Simpson, 1981). The tidal effects
considered in Simpson (1981) are neglected here due to the relatively weaker contribution in stratification in the shelf when
compared to the effects of rivers and winds. The two mixing terms are quantified as follows:

$\frac{d(PEA)}{dt} = \frac{\alpha g}{2c}Q - \delta k_a \rho_a W^3,$           (2)

where $Q$ is the rate of surface heat input, $\alpha$ is the volume expansion coefficient, $c$ is water specific heat capacity, $\delta$ is a
coefficient of wind mixing, $k_a$ is drag coefficient, $\rho_a$ is humid air density near the sea surface, and $W$ is the wind speed near
the sea surface. The first term on the right-hand side of Eq. (2) represents the rate of change of water stratification due to
surface heating, while the second term is the rate of working by wind stress contributing negatively to water stratification.
Therefore, the heat-induced change of PEA is proportional to surface heat input, which is,

$d(PEA)_{heat} \propto Q,$           (3)

The total net heat flux, a sum of net shortwave and net longwave radiation flux, is derived from the National Centers for
Environmental Prediction Climate Forecast System (NCEP) Reanalysis (CFSR) 6-hourly products (Saha et al., 2010; 2011) in
this study. The term Q is added to the candidate list of predictors and is denoted as $PEA_{heat}$ (heat-induced PEA changes) for
simplification (Figure 1a).

Daily variability of term ($\delta k_a \rho_a W^3$) is dominated by that of $W^3$, since the $\rho_a$ fluctuates much less than the $W^3$ on a daily
scale (Figure A2). We obtained the $\rho_a$ according to (Picard et al., 2008) :

$\rho_a = \frac{pM_d}{ZRT}\left[1 - x_v\left(1 - \frac{M_v}{M_d}\right)\right],$           (4)

where $p$ represents the absolute air pressure, $M_d$ (=28.96546 g mol$^{-1}$) is the molar mass of dry air, $M_v$ (=18.01528 g mol$^{-1}$) is
the molar mass of water vapor, $Z$ indicates compressibility, $R$ (=8.314472 J mol$^{-1}$ K$^{-1}$) is the molar gas constant, $T$ is
thermodynamic temperature, $x_v$ is the mole fraction of water vapor. We assumed that air parcels at the sea surface are ideal
gases ($Z = 1$) and are always saturated with water vapor. Thus, $x_v$ is a function of absolute air pressure ($p$) and saturation
vapor pressure of water ($p_{sat}$) and can be calculated as follows:

$$x_v = \frac{p_{sat}}{p},\tag{5}$$

According to the adjusted Tetens equation (Murray, 1967; Monteith and Unsworth, 2014), $p_{sat}$ (in Pa) can be estimated by:

$$p_{sat} = 611e^{\frac{17.27(T-237.3)}{T-T'}},\tag{6}$$

where $T' = 36$ K. Substitute Eqs. (5)–(6) to Eq. (4) with the assumption of $Z = 1$, we obtained air density as a function of both
air pressure and air temperature in the following:

$$\rho_a = \rho_a(T,p) = \frac{pM_d}{RT}\left[1 - \frac{611}{p}\left(1 - \frac{M_v}{M_d}\right)e^{\frac{17.27(T-237.3)}{T-T'}}\right],\tag{7}$$

The $\rho_a$ is then estimated using sea surface air pressure and air temperature 2 meters above the sea surface provided by NCEP
CFSR 6-hourly products. The correlation of daily $\rho_a W^3$ and $W^3$ (provided by NCEP CFSR 6-hourly products) is significantly
high as 0.9988 ($p<0.001$, Figure A2) emphasizing the importance of term $W^3$ in controlling the daily variability of wind-
induced PEA changes over the shelf. We, thus, approximated the relationship as:

$$d(PEA)_{wind} \propto W^3,\tag{8}$$

The term $W^3$ is introduced as another candidate predictor and is denoted as PEA$_{wind}$ (wind-induced PEA changes) for
simplification (Figure 1a).

## 2.1.2 Biochemical-related predictors

Sedimentary biochemical processes directly influence the bottom DO consumption rate. However, global forecast models such
as HYCOM do not cover biochemical parameters. Therefore, the biochemical-related term SOC needs to be replaced by an
alternative term (denoted as SOCalt). According to the SOC scheme (Eq. 9) stated in Part I, the biochemical features are
attributed to the sedimentary particulate organic nitrogen concentration (PONsed, derived from ROMS hindcasts). The total
nitrate+nitrite loads by the Mississippi River are used to represent the PONsed variability due to the long-term data supports.
The daily Mississippi River discharges at site 07374000 are updated daily by the U.S. Geological Survey (USGS) National
Water Information System (NWIS) since March 2004. The total nitrogen concentration at site 07374000 is provided and
updated daily by USGS since November 2011. Prior to 2011, nitrogen loads (at site 07374000) are provided monthly by USGS
and, in this study, are interpolated to daily intervals according to the corresponding monthly loads. Although phosphate and
silicate are another two limitation nutrients in the shelf, daily measurement are still not available for the Mississippi River.
Monthly total nitrate+nitrite loads, phosphate loads, and silicate loads by both the Mississippi River and the Atchafalaya River
are significantly correlated (Table A1). Therefore, the total nitrate+nitrite loads applied here can be interpreted as total nutrient
loads by both river systems. Due to lateral transports and vertical settling of particulate organic matter, a leading period should
be introduced to the time series of riverine nutrient loads. The optimal length of leading days is obtained by examining the
highest linear correlation of regionally averaged ROMS-hindcast SOC and SOCalt (Eq. (10)) and is calculated as 44 days
(R=0.7427, $p$<0.001, Figure A3a). The exponential term in Eqs. (9)–(10) estimates the temperature-dependent decomposition
rate of organic matter.
$SOC = PON_{sed} \cdot VP2N_0 \cdot e^{K_{P2N} \cdot T_b}$,  (9)
$SOCalt = $ Mississippi River inorganic nitrogen loads (led by 44 days) $\cdot e^{0.0693 T_b}$,  (10)

$VP2N_0$ is a constant representing the decomposition rates of sedimentary particulate organic nitrogen, $PON_{sed}$, at 0 ºC. $K_{P2N}$
is a constant (0.0693 ºC$^{-1}$) indicating temperature coefficients for decomposition of $PON_{sed}$. $T_b$ is bottom water temperature
(in °C). The Q10 (= 2 given the above chosen coefficients; van't Hoff and Lehfeldt, 1899; Reyes et al., 2008) assumption is
applied to mimic the aerobic decomposition rate of $PON_{sed}$. Along with SOCalt, the temperature-dependent decomposition
rate $e^{0.0693 \cdot T_b}$ is also considered as a candidate predictor in statistical models and is denoted as DCP$_{Temp}$ for simplification.
**2.1.3 HA estimation**
As listed in Table 1, six candidate predictors are considered in the statistical models including four stratification-related
variables (PEA, SSS, PEA$_{heat}$, and PEA$_{wind}$) and two bottom biochemical variables (SOCalt and DCP$_{Temp}$). The correlation
coefficient matrix (Figure 1a) indicates that multicollinearity may become a problem in regression models since linear
correlations among some predictors are significantly high, e.g., 0.74 ($p$<0.001) between PEA and SOCalt, and -0.87 ($p$<0.001)
between PEA and SSS. The multicollinearity can harm the assumption that predictors are independent. It can lead to difficulties
in individual coefficients test and numerical instability (Siegel and Wagner, 2022). The frequency distribution of HA (Figure
1b) illustrates that the response variable is highly right-skewed with ~42 % of samples (2,081 out of 4,943) being exactly zero.
The HA is estimated by the number of hypoxia cells (ROMS computational cells reaching hypoxic conditions) times a nearly
constant value (area of the computational cell), which is 25.56 $\pm$ 0.17 km$^2$ (mean $\pm$ 1SD). Thus, the HA can be estimated by
the number of grid cells when the Poisson and negative binomial regression models are applied. However, the great portion of
zero samples leads to overdispersion (magnitude of variance $\gg$ magnitude of mean, i.e., 45,730,441 $\gg$ 4,507) and zero-
inflated problems (Lambert, 1992). The overdispersion issue violates the mean-variance equality assumption employed in
regular Poisson regression models, while zero-inflated problems can weaken the model performances.
**Table 1. Description of daily response variable and candidate predictors. The data cover a time range from 1 January**
**2007 to 26 August 2020. Prescribed min and max are used for min-max normalization.**

| Variables [units] | Description | Min | Median | Mean | Max | Prescribed (Min:Max) |
|---|---|---|---|---|---|---|
| HA [km$^2$] | Hypoxic area (when bottom dissolved oxygen $< 2$ mg L$^{-1}$) | 0 | 1,137 | 4,507 | 34,097 | Non-normalized |
| PEA [J m$^{-3}$] | Potential energy anomaly measuring the water stratification | 3.3 | 35.6 | 47.2 | 187.9 | (0:200) |
| SSS [non-dim] | Sea surface salinity | 20.0 | 30.8 | 30.4 | 33.9 | (0:40) |
| PEA$_{heat}$ [W m$^{-3}$] | =Q, an approximation of surface heat-induced water stratification | -54.4 | 151.9 | 142.7 | 261.3 | (-60:300) |
| PEA$_{wind}$ [m$^3$ s$^{-3}$] | =W$^3$, an approximation of water stratification changes due to wind mixing | 0.5 | 164.7 | 296.1 | 7013.2 | (0:7,100) |
| SOCalt [mmol s$^{-1}$] | An alternative term for sediment oxygen consumption. | 789,319 | 10,423,383 | 13,377,287 | 41,984,069 | (770,000:43,000,000) |
| DCP$_{Temp}$[non-dim] | $= e^{0.0693 \cdot T_b}$, temperature-dependent decomposition rate of organic matter | 2.6 | 5.1 | 5.2 | 8.0 | (0:10) |


## 2.2 Data pre-processes

We first spatially averaged ROMS-derived predictors (daily) over the LaTex Shelf (color-shaded area in Figure A1b), then
applied the min-max normalization (Eq. (11)) to the one-dimensional time series. Predictive models can be beneficial from the
min-max normalization when applying to a new dataset since the method guarantees that the normalized predictors from
different datasets range from 0 to 1 as the minimum and maximum values are prescribed. Note that the response is not
normalized.

$X_{nor} = \frac{X_{org} - Min_{prescribed}}{(Max_{prescribed} - Min_{prescribed})},$          (11)

where $X_{nor}$, $X_{org}$, $Min_{prescribed}$, and $Max_{prescribed}$ represent normalized value, original value, prescribed minimum, and
prescribed maximum, respectively. The daily samples are then split into a training set (for model construction) accounting for
80 % of the total samples and a test set (for assessment of model performances) accounting for the remaining 20 %. To maintain
the HA distribution in both sets, a random resampling method is applied in different HA intervals individually. For example,
80 % of samples with HA=0 are chosen randomly for the training set out of all daily samples with HA=0, while the rest of the
samples with HA=0 are grouped into the test set. The HA=0 is the first interval to which the resampling process is applied,
while the remaining samples are split at intervals of 5,000 km². However, the distribution of HA from each year is similar with
a right-skewed structure and numerous zero values. Thus, even through random processes, both the training and test sets
contain samples from each year including samples with non-peak and peak HA. This splitting method increases the model
applicability and provides a comprehensive assessment of prediction performances on both non-peak and peak HA.

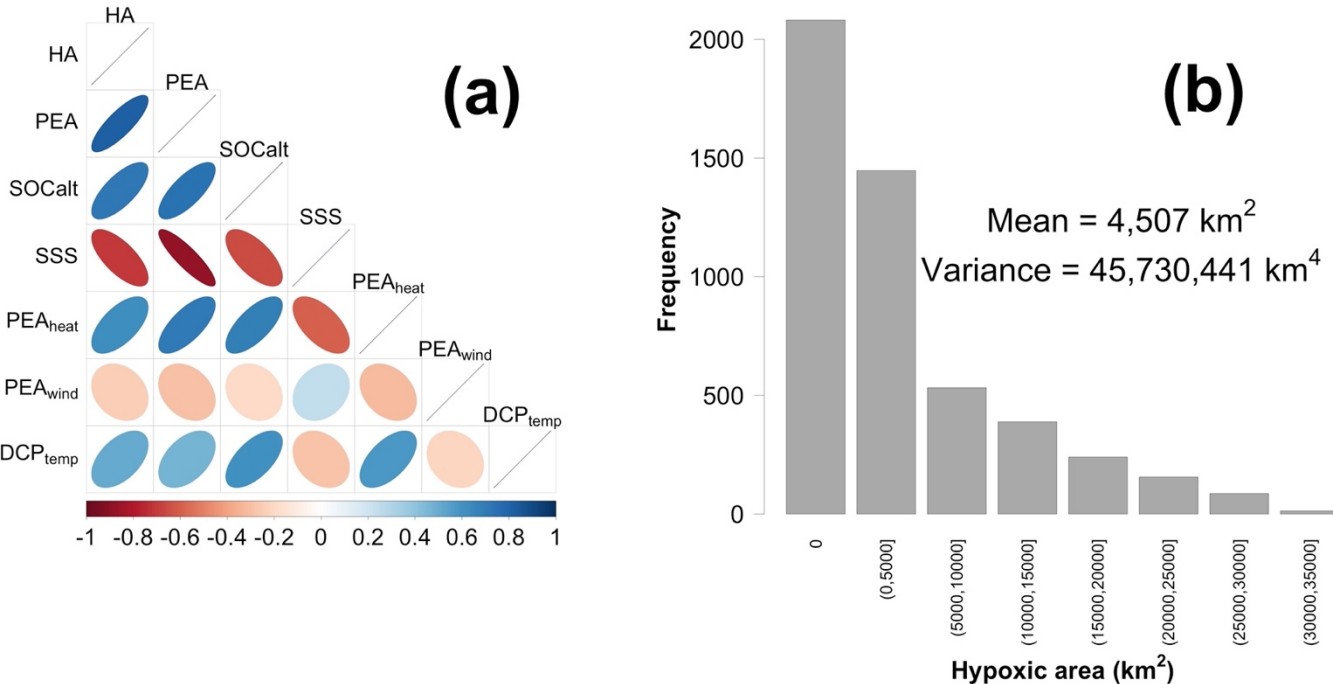


**Figure 1. (a) A correlation coefficient matrix of the response variable and candidate predictors, and (b) the frequency distribution**
**of HA. Data are provided daily from 1 January 2007 to 26 August 2020.**
**2.3 Model skill assessment**
The $R^2$, root-mean-square error (RMSE), mean absolute percentage bias (MAPB), and scatter index (SI; Zambresky, 1989)
are used to assess the model performances in HA predictions. The SI is a normalized measure of error or a relative percentage
of expected error with respect to the mean observation. The calculations of the statistics are given in Eqs. (12) – (15).
$R^2 = 1 - \frac{\sum_{i=1}^{N}(P_i - O_i)^2}{\sum_{i=1}^{N}(P_i - \bar{O})^2}$         (12)

$RMSE = \sqrt{\frac{\sum_{i=1}^{N}(P_i - O_i)^2}{N}}$         (13)

$MAPB = \frac{1}{N}\sum_{i=1}^{N}\left|\frac{P_i - O_i}{O_i}\right| \times 100\%$         (14)

$SI = \frac{RMSE}{\bar{O}} \times 100\%$         (15)
where $P_i$ and $O_i$ represent the $i$th record of prediction and observation (or hindcast), while $\bar{O}$ represents the average of all
observed (or hindcast) records.
**3 Model construction and results**
**3.1 Model built-up process**
Several regression models are explored using the statistical programming language R. To find the "best" model balancing both
model interpretability and prediction performance, a procedure is conducted for model selection (Figure 2) and is summarized
below. (1) Choose a regression model. (2) Apply an exhaustive best-subset searching approach to the chosen model. Models
with possible combinations of candidate predictors from the ROMS training set are built. A 10-fold cross-validation (CV)
method is applied to each model yielding 10 RMSEs and 1 corresponding mean. The candidate predictors of PEA and SOCalt
are forced into each subset. Thus, the number of fitted models with a subset size of k is $C(6-2, k-2) = \frac{4!}{(6-k)!(k-2)!}$ , $2 \leq$
$k \leq 6$ (the total number of candidate predictors is 6). The optimal subset of this size is found as the one with the lowest mean
CV RMSE among these models. The best subset is then obtained by comparing mean CV RMSEs of the optimal subsets of
different sizes. (3) Steps (1)–(2) are repeated for the selected M candidate regression models. (4) Prediction performances of
different models with the corresponding best subsets are assessed by the 10-fold CV RMSEs and Bootstrap (1,000 iterations)
aggregating (i.e., Bagging) ensemble algorithms. The Bagging method builds the given model N (=1,000) times during each
of which the given model is trained using different samples chosen randomly and repeatedly from the ROMS training set and
is executed for HA prediction using samples in the ROMS test set. The ensemble means and ensemble 95 % prediction intervals
(PIs) of forecast HA are given according to the prediction results in the 1,000 iterations. The best model (Model X in Figure
2) is chosen according to the comparisons of the 10-fold CV RMSEs and the Bagging results.

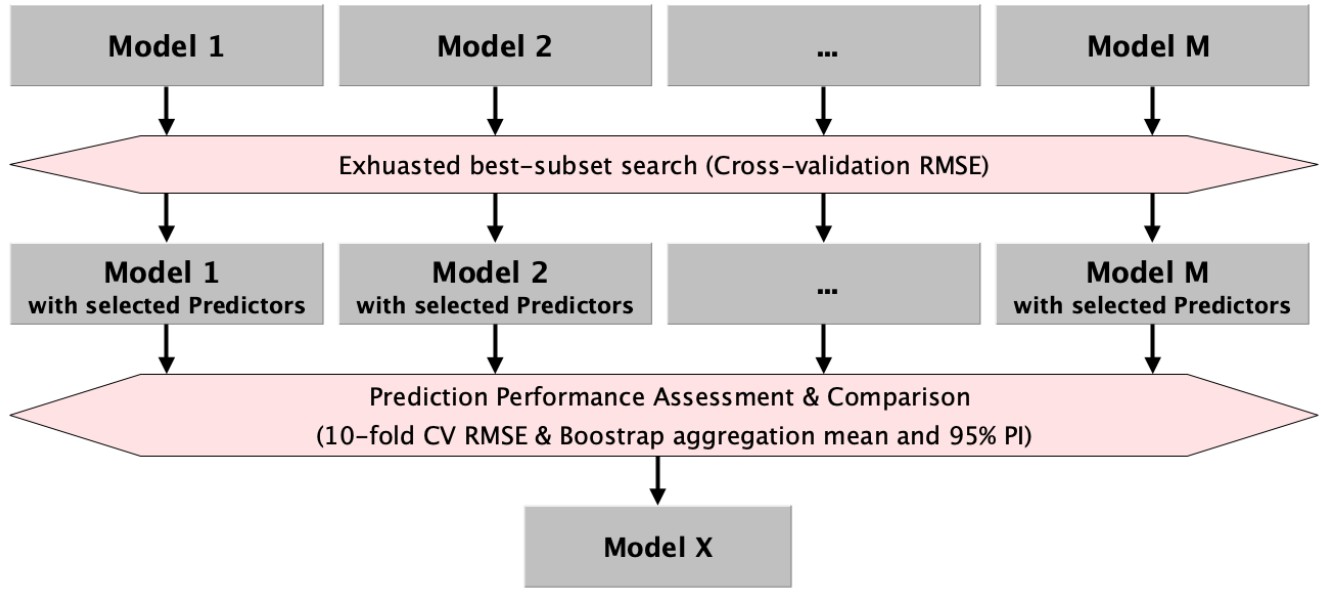

**Figure 2. A flow chart of building up regression models.**
**3.2 Generalized linear models (GLMs)**
**3.2.1 Regular GLMs and zero-inflated GLMs**
The response variable can be treated as count data. Regular Poisson (function glm in R package "stats" version 3.6.2), quasi-
Poisson (function glm in R package "stats" version 3.6.2), and negative binomial (function glm.nb in R package "MASS"
version 7.3-54; Venables and Ripley, 2002) GLMs are explored in this section. The latter two GLMs are known for solving
overdispersion problems by relaxing the mean-variance equality assumption. These GLMs make use of a natural log link
function. Thus, a natural logarithm of the area of a single ROMS cell ($\sim$ 25.56 km$^2$) is added to the models as an offset term
(an additional intercept term).

In addition, the overdispersion issue can result from the great percentage ($\sim$42 %) of zero values in the response variable
(Figure 1b). Zero-inflated GLMs (using function zeroinfl in R package "pscl" version 1.5.5; Jackman, 2020; Zeileis et al.,
2008) are developed for dealing with response variables of this kind. Rather than resetting dispersion parameters, a zero-
inflated count model is a two-component mixture model blending a count model and a zero-excess model. The count model is
usually a Poisson or negative binomial GLM (with log link), while the zero-excess model is a binomial GLM (with logit link
in this study) estimating the probability of zero inflation. An offset term of log (25.56) is also introduced into the count model.
Instead of applying the best-subset searching to the count and zero-excess models simultaneously, in this study, the searching
is conducted respectively for these two models to reduce the demands of computational resources. The best subset of the zero-
excess model (binomial GLM) is given first. The best subset of the count model (Poisson or negative binomial GLMs) is then
provided blending the zero-excess model with the corresponding selected best subset fixed.

However, it is hard to determine whether a given zero value of HA is excessive, instead, it is relatively easy to model hypoxia
occurrence assuming that all the zero values are excessive. A new binary response, hypoxia, stated in Eq. (16) is introduced
for modeling hypoxia occurrence using regular binomial GLMs (function glm in R package "stats" version 3.6.2). The hypoxia
is equal to 0 when HA is 0 (no hypoxia), otherwise, is equal to 1. The optimal model selected three predictors: PEA, SOCalt,
and $DCP_{Temp}$ (Figure 3b).

$hypoxia = \begin{cases} 0, & no\ hypoxia \\ 1, & hypoxia\ occurs \end{cases}$,     (16)

**3.2.2 Performance of GLMs**
The zero-inflated Poisson GLM serves as the best GLM in terms of prediction performances since it has the lowest mean CV
RMSE (Figure 3a) among the five candidates GLMs. The relaxation of the mean-variance equality assumption by the negative
binomial GLM and the quasi-Poisson GLM does not guarantee salient improvement of performances when comparing their
CV RMSEs to those of regular Poisson GLM. The zero-inflated negative binomial GLM yields similar performances to the
three regular GLMs. The mean CV RMSEs of zero-inflated Poisson GLM hit the trough (3,573 $km^2$) at the size of four.
However, the greatest drop of RMSEs (3,586 $km^2$) occurs at the size of three beyond which the RMSEs remain stable. It is
worth considering a model with fewer predictors satisfying model interpretability. Thus, the best zero-inflated Poisson GLM
accounts for three predictors (PEA, SOCalt, and $DCP_{Temp}$) in the count model and three predictors (PEA, SOCalt, and $DCP_{Temp}$)
in the zero-excess model. As indicated in the correlation matrix (Figure 1a), the robustness of a model can be impaired by
multicollinearity which can be estimated by variance inflation factors (VIFs). VIFs among the selected predictors are 2.15,
2.70, and 1.59 for PEA, SOCalt, and $DCP_{Temp}$, respectively. The VIFs are all less than 5 suggesting that both the count and the
zero-excess models with these predictors involved are merely violated by multicollinearity. For simplicity, the best zero-
inflated Poisson GLM is symbolized as GLMzip3.

The Bagging ensemble method is implemented to estimate the prediction performance of GLMzip3 (Figure 4a). It is noted
that the training set and test set are resampled according to different HA intervals. Since the distributions of HA in each year
are similar (see Section 2.2), HA in both training and test set contains observations of peak and non-peak values in each year.
Therefore, samples shown in Figure 4 are listed sequentially in the time dimension from 2007 to 2020 but are not necessarily
evenly distributed. The listed samples should not be regarded as time series. The Bagging means of predicted HA provides an
RMSE of 3,614 $km^2$ and an $R^2$ of 0.7214 against the ROMS hindcasts. The Bagging 95 % PIs are restricted within a narrow
range with a slight increase at the predicted peaks. Within different ranges of hindcast HA, the MAPB between predicted and
hindcast HA ranges from 29 % to 38 % with an average of 33 % (Table 2). Particularly, the GLMzip3 produces the lowest
bias (29 %) for the hindcast HA ≥ 30,000 $km^2$. The results suggest that GLMzip3 is capable of providing not only accurate
but also stable HA forecasts. Nevertheless, we noted salient overestimations (e.g., peaks around samples 30, 481, and 901) and
underestimations (e.g., peaks around samples 181, 390, and 826) at some peaks. Instead of the prediction performance at non-
peak HA, here we focused more on the forecasts at HA peaks which impose more threats to the shelf ecosystem. In section
3.3, GAMs are investigated with an expectation of further improvements in peak predictions by considering non-parametric
or non-linear effects of the predictors.

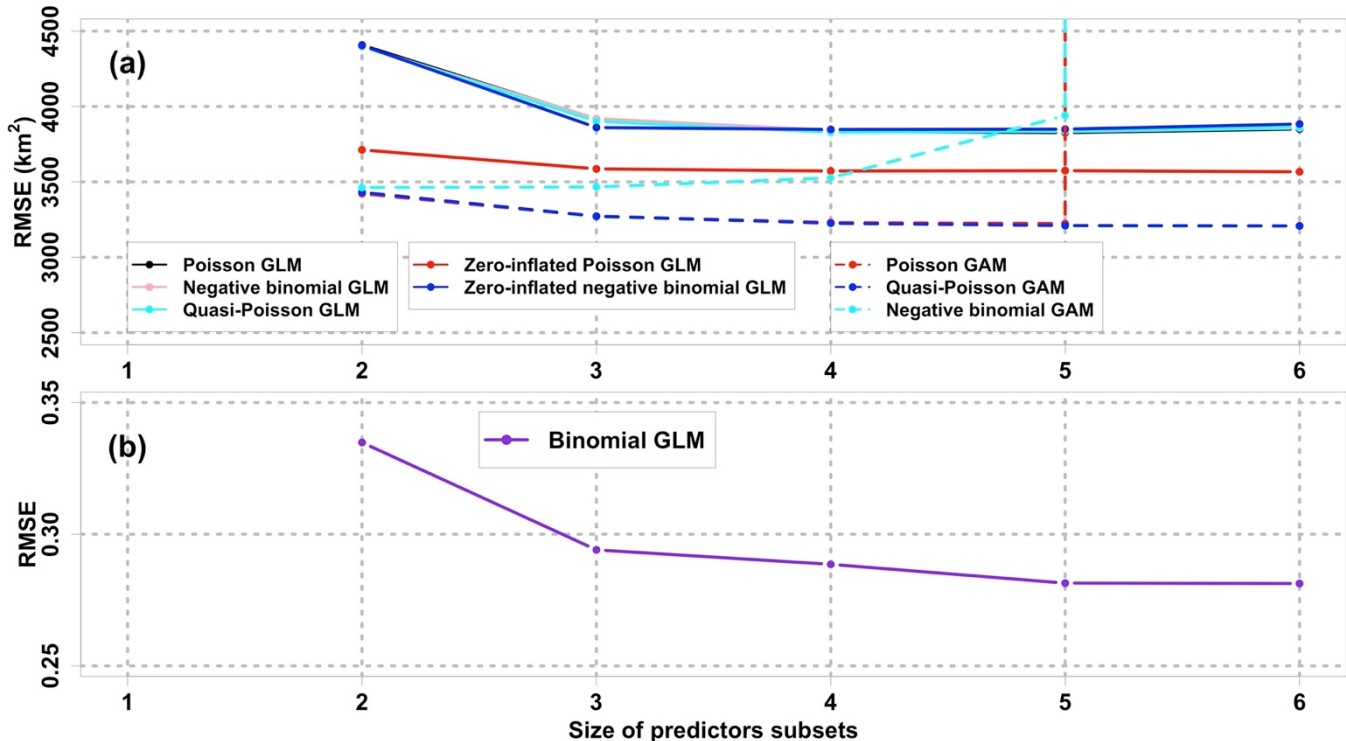


**Figure 3. Comparisons of mean 10-fold CV RMSEs among different regression models with various sizes of predictors subsets. The**
**response variable in (b) binomial GLM and (a) other models is hypoxia occurrence (hypoxia) and hypoxic area (HA), respectively.**
**Note that the CV RMSEs of negative binomial GAM and Poisson GAM with the size of six are out of the range shown. CV RMSE**
**curves of the Poisson GLM, negative binomial GLM, and quasi-Poisson GLM overlap, while those of Poisson GAM and quasi-**
**Poisson GAM overlap when size ≤ 5. The minimum size of predictor subsets is two since PEA and SOCalt are forced into every**
**subset.**

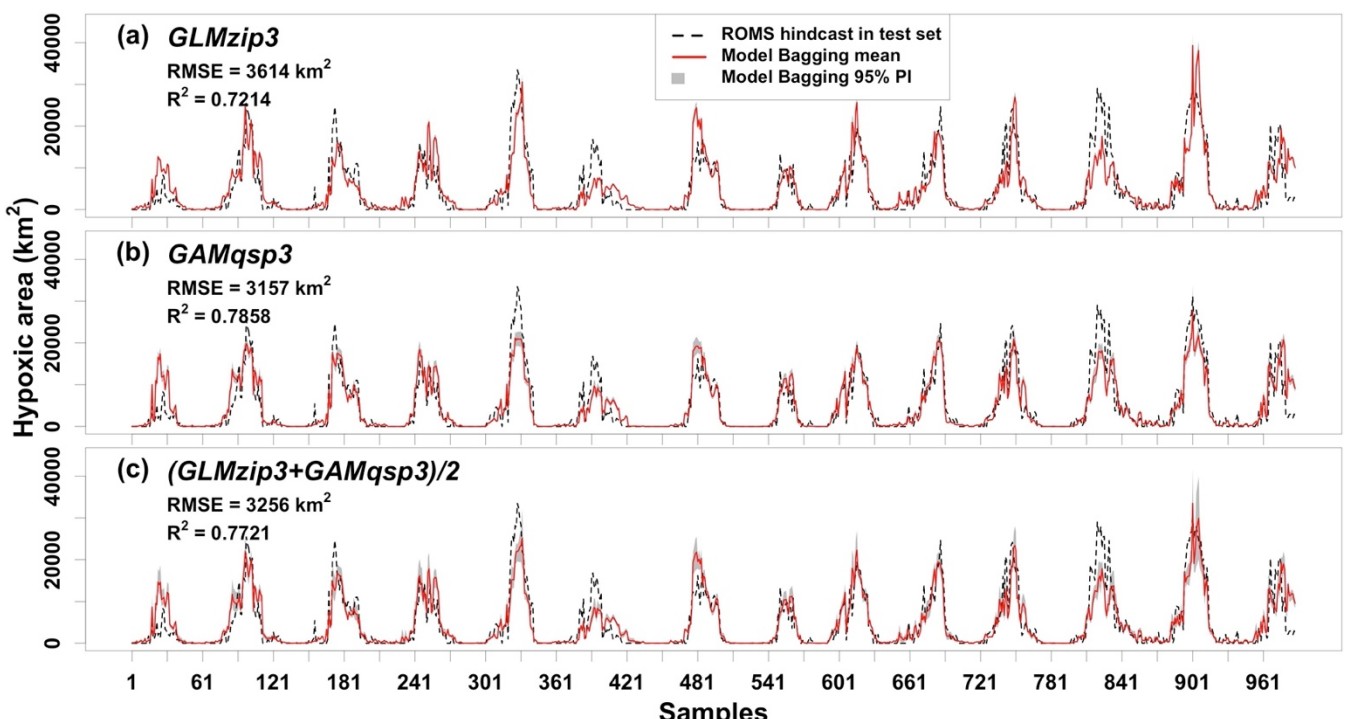

**Figure 4. Comparisons of model predicted HA and ROMS-hindcast HA in the test set. RMSEs and R²s are derived between model Bagging mean and ROMS-hindcast HA.**

Table 2 Mean absolute percentage bias between predicted and hindcast HA in the test set within different ranges of hindcast HA. The mean bias when hindcast HA $< 5,000$ km² is not shown since the prediction accuracy at high HA ranges is a more important feature of HA prediction models. The threshold of 5,000 km² is chosen because it is the goal HA set by the Action Plan (Mississippi River/Gulf of Mexico Watershed Nutrient Task Force, 2001; 2008). HA above this threshold is more worthy of attention.

| Hindcast HA range (km²) | GLMzip3 | GAMqsp3 | Ensemble |
|---|---|---|---|
| [5000, 10000] | 38 | 40 | 36 |
| [10000, 20000] | 32 | 25 | 28 |
| [20000, 30000] | 34 | 26 | 28 |
| $\geq 30000$ | 29 | 28 | 25 |
| Average | 33 | 30 | 29 |

### 3.2.3 Model interpretation for GLMzip3

We applied the complete ROMS training set to the model construction of GLMzip3. Coefficients for PEA, SOCalt, and DCP$_{\text{Temp}}$ (Table 3) are all found significantly positive ($p<0.001$) in the count model, while coefficients for these predictors are significantly negative ($p<0.001$) in the zero-excess model. The count model simulates the HA while the zero-excess model estimates the probability of HA being zero. Higher PEA is consistent with stronger water stratification, while higher SOCalt

and DCP$_{Temp}$ are both corresponding to higher sediment oxygen consumption. Therefore, there is no surprise that higher PEA, SOCalt, and DCO$_{Temp}$ are related to greater HA and higher hypoxia occurrence or lower probability of HA being zero. Results indicate that the GLMzip3 essentially builds up reasonable relationships between the response and predictors variables with a high agreement with physical and biochemical mechanisms. Since the ranges of normalized predictors are from 0 to1, comparisons of regression coefficients indicate that effects of PEA (2.8037 in the count model and -10.4439 in the zero-excess model, same hereafter) are considered more important than SOCalt (0.9057 and -7.3100) and DCP$_{Temp}$ (0.8425 and -95698). The result is consistent with the findings of previous studies which emphasized that the physical impacts are stronger than the biological impacts on HA estimates (Yu et al., 2015; Mattern et al., 2013).

**Table 3. Regression coefficients of GLMzip3.**

| Count model coefficients (Poisson with log link): | | | | Zero-excess model coefficients (binomial with logit link): | | | |
|---|---|---|---|---|---|---|---|
| | Estimate | Std. Error | z value | Pr (> \|z\|) | | Estimate | Std. Error | z value | Pr (> \|z\|) |
| Intercept | 3.6397 | 0.0017 | 2120.5 | <2E-16*** | Intercept | 7.7641 | 0.2761 | 28.12 | <2E-16*** |
| PEA | 2.8037 | 0.0014 | 1984.6 | <2E-16*** | PEA | -10.4439 | 0.6794 | -15.37 | <2E-16*** |
| SOCalt | 0.9057 | 0.0014 | 639.6 | <2E-16*** | SOCalt | -7.3100 | 0.5714 | -12.79 | <2E-16*** |
| DCP$_{Temp}$ | 0.8425 | 0.0029 | 287.7 | <2E-16*** | DCP$_{Temp}$ | -9.5698 | 0.4611 | -20.75 | <2E-16*** |

Significance codes: 0 (***) 0.001 (**) 0.01 (*)

Log-likelihood: -2.675E6 on 8 degrees of freedom

## 3.3 Generalized additive models (GAMs) and the ensemble model

GAMs are explored with an expectation of improving prediction performance in HA peaks by introducing non-parametric effects of predictors. Using function "gam" in R package "mgcv" (version 1.8-36; Wood, 2011) with smooth functions as pure thin plate regression splines (degree of freedom=9; Wood, 2003), three GAMs are studied and compared, i.e., Poisson GAM, quasi-Poisson GAM, and negative binomial GAM. Following the same procedure in GLM exploration, the best subset searching approach is applied to the GAMs first. Although mean 10-fold CV RMSEs for the Poisson and quasi-Poisson GAMs (Figure 3a) exhibit insignificant differences at sizes from two to five, the CV RMSEs for the former increase dramatically at a size of six, which indicates that the model stability decreases with sizes. The negative binomial GAM has the greatest mean CV RMSEs among the GAMs studied and has an extremely high mean CV RMSE at the size of six. The quasi-Poisson GAM is considered the best GAM among the three. Although the mean CV RMSEs for the quasi-Poisson GAM reach the lowest at the size of six, the best size is considered as three (including PEA, SOCalt, and DCP$_{Temp}$) at which CV RMSEs exhibit the most saline decline, and beyond which mean CV RMSEs stabilize around 3,200 km$^2$. The quasi-Poisson GAM with three predictors involved is symbolized as GAMqsp3.

Component plots of the GAMqsp3 (Figure 5) imply that HA generally increases as the chosen predictors increase. Note that
the summation of all smooth function terms contributes directly to the log of fitted HA. Such results agree with those found
by model GLMzip3. However, the component plots provide more detailed information about the rate of changes in HA. The
effective degrees of freedom range from 6.79 to 8.90 indicating strong non-linear effects of the predictors on the variability of
HA. The HA is more sensitive to the predictors in the low-value ranges but becomes nearly stable in the medium- and high-
value ranges of predictors. This implies that bottom hypoxia develops rapidly in early summer when water stratification and
sediment oxygen demand start to increase. On the other hand, the smooth functions of SOCalt and DCP$_{Temp}$ have a sharper
slope than that of PEA at the low-value range. It suggests that at the first stage of hypoxia development in late spring and early
summer, sedimentary biochemical processes contribute more than water stratification. The bottom hypoxic water further
extends with a much lower expansion speed as the stratification and SOCalt further intensify. Nevertheless, the smooth function
of PEA is slightly greater also with a more acute slope than those found for SOCalt and DCP$_{Temp}$ in the medium- and high-
value regimes of the predictors. It indicates that the HA variability is more related to the hydrodynamic changes in the shelf
than the biochemical effects during mid-summers. The result is consistent with the findings by Yu et al., (2015) and Mattern
et al. (2013). The GAMqsp3 model provides reasonable interpretations on the hypoxic area mechanisms.

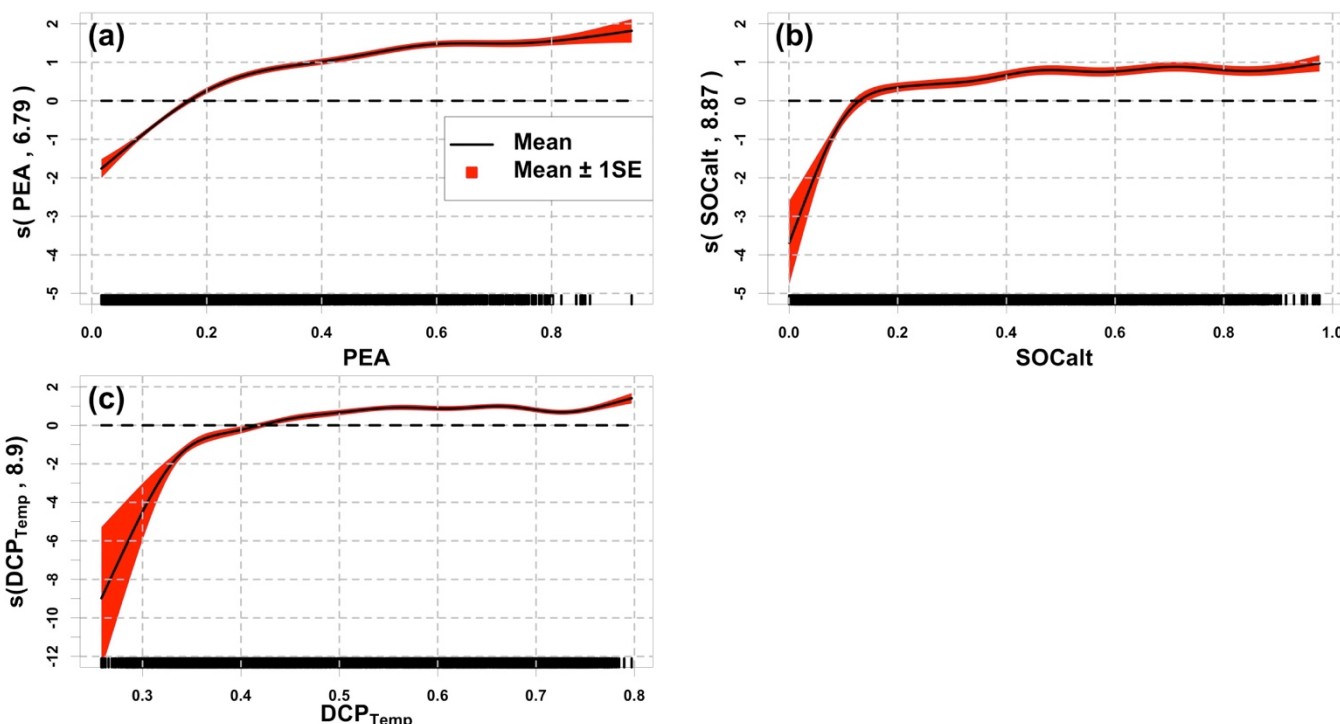

**Figure 5. Component plots of model GAMqsp3. Solid black lines represent the mean of the smooth function, while the red area**
**denotes the range of mean ± 1SE. Numbers in brackets represent effective degrees of freedom for the corresponding smooth terms.**
**Black bars at the x axis indicate the density of corresponding normalized predictors. Dashed black lines are straight lines of zero**
**along the predictor domains.**
The prediction performance of GAMqsp3 is estimated using the Bagging ensemble method (Figure 4b). The RMSE and $R^2$
between the Bagging mean and ROMS-hindcast HA is 3,157 $km^2$ and 0.7858, respectively. They are 13 % lower and 9 %
higher than the corresponding statistics found for the GLMzip3, respectively. MAPB between GAMqsp3 predicted and
hindcast HA ranges from 25 % to 40 % with an average of 30 % (Table 2). Such statistics are generally lower than those found
in GLMzip3. Results suggest that GAMqsp3 outcompetes GLMzip3 in terms of overall performance. However, GAMqsp3
tends to underestimate HA peaks (like those seen at peaks around samples 750 and 901) some of which are overestimated by
the GLMzip3. Therefore, instead of determining the best model out of the two, ensemble HA predictions blending efforts of
both GLMzip3 and GAMqsp3 are carried out with an expectation to improve model performance in the peak forecast. We
assumed that the contributions of GLMzip3 and GAMqsp3 are equally weighted and thus averaged the predicted HA by
GLMzip3 and GAMqsp3 and calculated the 95 % PIs given the Bagging results of these models (Figure 4c). As expected, the
overall performance of the ensemble forecast is somewhere between the performance of GLMzip3 and GAMqsp3 with an
RMSE of 3,256 $km^2$ and an $R^2$ of 0.7721. However, some HA peak events (like peaks around samples 750 and 901) which are
overestimated by GLMzip3 but are underestimated by GAMqsp3 are accurately predicted by the ensemble approach. MAPB
also indicates an increase in peak prediction performance by the ensemble model. The statistic is within a range of 25 % to 36
% with an average of 29 %. At extreme peaks (hindcast HA $\geq$ 30,000 $km^2$), compared to the MAPB by GLMzip3 (29 %) and
by GAMqsp3 (28 %), the statistic decreases to 25 % by the ensemble model. The ensemble model provides a higher accuracy
in peak forecast given minor sacrifices in overall performance.
**3.4 Application to Global Forecast Products (HYCOM)**
The power of the prediction model relies on the availability of the forecast of predictors. In this section, we discuss the model's
transferability using an independent global ocean product. The Global Ocean Forecasting System (GOFS) 3.1 provides global
daily analysis products and an eight-day forecast in a daily interval with a horizontal resolution of 1/12 °. The products
(hereafter referred to as HYCOM-derived products) are derived by a 41-layer HYCOM global model (Bleck and Boudra, 1981;
Bleck, 2002) with data assimilated via the Navy Coupled Ocean Data Assimilation (NCODA) system (Cummings, 2005;
Cummings and Smedstad, 2013). The Mississippi River total nitrate+nitrite loadings are provided by USGS NWIS as described
in section 2.1.2. Daily HYCOM-derived hydrodynamics and USGS river nitrogen loads from 1 January 2007 to 26 August
2020 are used to reconstruct predictors of PEA, SOCalt, and $DCP_{Temp}$. Relationships of ROMS-derived and HYCOM-derived
predictors are examined in Figure 6. The magnitudes of HYCOM-derived SOCalt and $DCP_{Temp}$ match up with the
corresponding ROMS-derived predictors, respectively, although HYCOM-derived predictors are found slightly greater.
Simple linear regression for these predictors illustrates that the linear relationships between the ROMS and HYCOM products
are significant with the $R^2$ ranging from 0.94 to 0.96. The intercept terms are at least one-order smaller than the magnitudes of
corresponding predictors. Therefore, the HYCOM global products are deemed to agree with the ROMS hindcasts for SOCalt
and $DCP_{Temp}$. Nevertheless, the magnitude of HYCOM-derived PEA is found much lower than the ROMS-derived PEA
(Figure 6a). Simple linear regression indicates a significant linear relationship between the natural log transformation of PEA
from the two datasets ($R^2$=0.66).

At land-sea interfaces, the HYCOM global model is forced by monthly riverine discharges, which weaken the model
performance in coastal regions. The hydrodynamics in the LaTex Shelf is highly affected by the freshwater and momentum
from the Mississippi and the Atchafalaya Rivers. Monthly river forcings in HYCOM are essentially weaker than daily forcings
used in our ROMS setups and can result in a less stratified water column (i.e., lower PEA). Therefore, it is necessary to scale
the magnitude of HYCOM-derived PEA to that of the ROMS hindcast. It can be achieved by using the natural log
transformation and simple linear regression as discussed. We then adjusted HYCOM-derived PEA but kept the HYCOM-
derived SOCalt and $DCP_{Temp}$ unchanged before the application of the ensemble model.

The Bagging approach is implemented again to assess the performances of the ensemble model. During each iteration
(N=1,000), the GLMzip3 and GAMqsp3 are trained using the ROMS training set and then applied to the adjusted HYCOM-
derived predictors for HA prediction from 1 January 2012 to 26 August 2020 (Figure 7a). The ensemble method provides
averages and 95 % PIs of predicted HA blending Bagging results by GLMzip3 and GAMqsp3. Compared to observed HA by
mid-summer Shelf-wide cruises, the ensemble model fails in the summers of 2013, 2014, 2017, and 2018, but provides accurate
predictions in other summers. The width of 95 % PI is larger during high HA periods suggesting less stability in the HA peak
forecast. The overall performance is barely acceptable with an $R^2$ of 0.4242, an RMSE of 5,088 $km^2$, and a SI of 38%. The
bias against the observations can be ascribed to the HYCOM's failures in reproducing the shelf hydrodynamics, although
HYCOM-derived predictors are adjusted before being applied to the model (Figure 6a). We noticed that among the three
variables, HYCOM-derived PEA exhibits the largest deviation from that generated by ROMS. We then applied the model
using ROMS-derived PEA, HYCOM-derived SOCalt, and HYCOM-derived $DCP_{Temp}$ (Figure 7b). The performance of the
ensemble model was largely enhanced with a higher $R^2$ (0.9255), a much lower RMSE (3,751 $km^2$), and a lower SI (28%)
compared to that using pure HYCOM products. These results indicate that the ensemble model can produce a highly accurate
prediction for HA summer peaks once water stratification is well resolved. Instead of using monthly river forcings, the
HYCOM model may possibly resolve the shelf hydrodynamics by utilizing daily river discharges of the Mississippi and the
Atchafalaya Rivers.

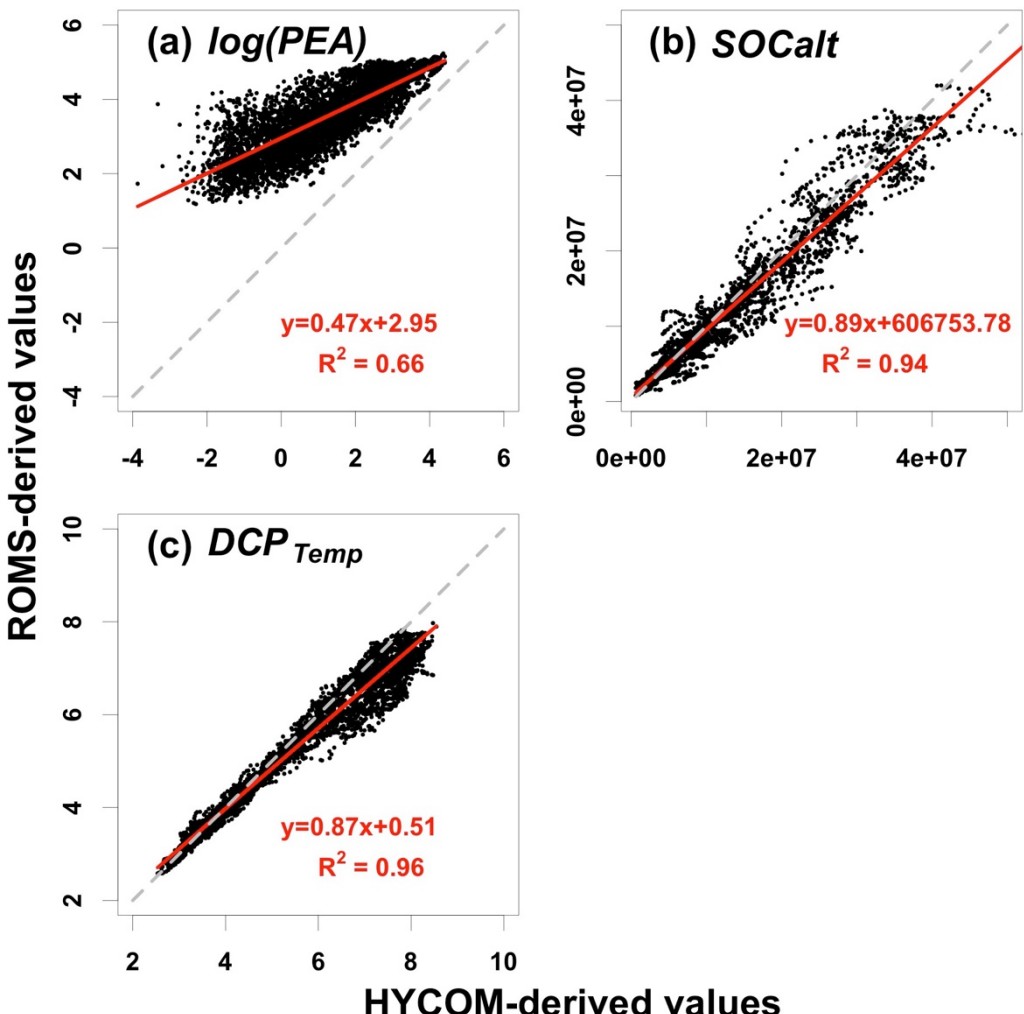


Figure 6. Scatter plots of (a) log(PEA) (unit: log $(J\,m^{-3})$), (b) SOCalt (unit: mmol $s^{-1}$), and (c) DCP$_{Temp}$ (unit: 1) between ROMS
and HYCOM simulations. Note that the solid red lines represent linear regression lines, while the dashed grey lines are diagonals
with a slope of 1 and an intercept of 0. Daily data compared are from 2007 to 2020.

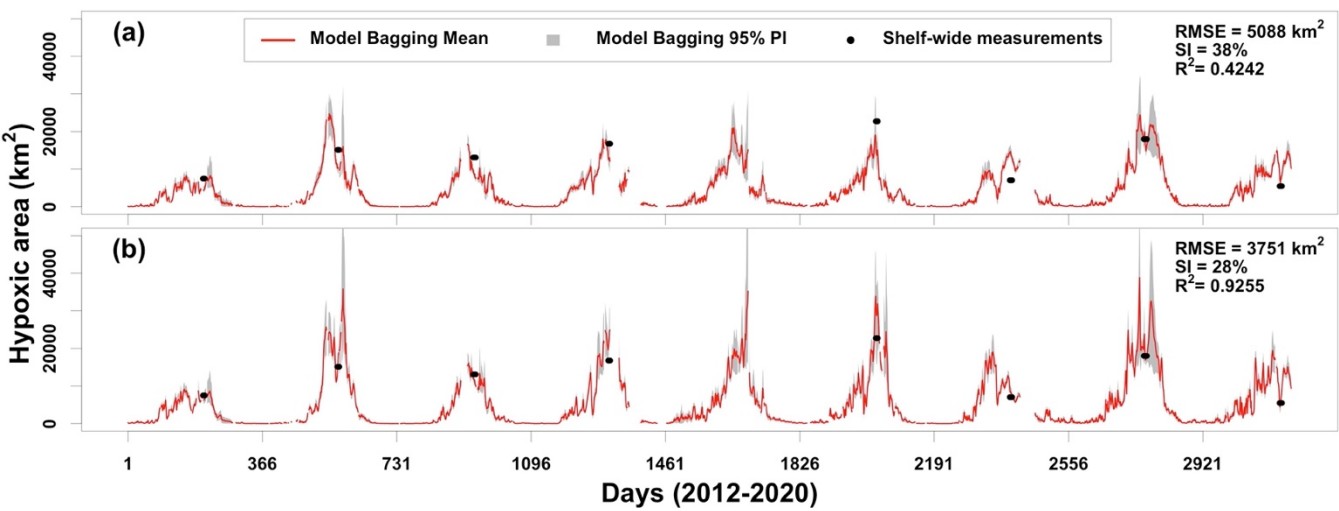


**Figure 7. Comparisons of daily predicted HA by ensemble model ((GLMzip3+GAMqsp3)/2) when applied to adjusted HYCOM products and Shelf-wide measurements from 2012 to 2020. Model results shown in (a) are predicted using pure HYCOM-derived products (i.e., PEA, SOCalt, and DCP$_{Temp}$), while those in (b) are predicted by ROMS-derived PEA, HYCOM-derived SOCalt, and HYCOM-derived DCP$_{Temp}$. Discontinuity of the predictions is due to the lack of riverine nitrate+nitrite records at site USGS 07374000 in the Mississippi River.**

## 4 Discussion

### 4.1 Model performance evaluation

To further assess the robustness of our model, we reviewed a suite of existing forecast models that are transitioned operationally (in early June) to the NOAA ensemble forecast for each summer (data sources are listed in Table 4). Using the ROMS-derived predictors, daily HA predictions during the Shelf-wide cruises periods are averaged for each summer from 2012 to 2020 and are compared to the cruise observations. As shown in Figure 8a, our model predictions fit well with the Shelf-wide observation for summers with or without strong windy events prior to the cruises. Other seasonal forecast models have similar performances to our model in fair-weather summers (i.e., 2012, 2014, 2015, and 2017) but fail to produce an accurate forecast for several summers with strong wind conditions (i.e., 2018 and 2020). Percentage differences between predictions and observations (Figure 8b) also emphasize the superiority of our model with the percentages ranging from -24 % to 7 % for fair-weather summers and from 7 % to 35 % for summers with strong wind or storms. All models underestimate or overestimate observed HA in fair weather summers, but overestimate HA in windy summers. Our model provides the most accurate overall performance with the highest $R^2$ (0.9200, N=8), the lowest RMSE (2,005 km$^2$, N=8), the lowest SI (15 %, N=8), and the lowest MAPB (18 %, N=8) among all models (Table 4). The multiple linear regression model developed by Forrest et al. (2011) provides the second optimal prediction. For fair-weather summers, the NOAA ensemble predictions produce the best estimation of the observed HA with a MAPB of 9 % (N=4), while our model results rank the second (15 %, N=4). However,

our model performs the best in windy summers with a MAPB of 18 % (N=4), while other models produce a MAPB from 33 %
to 74 %.

Models developed by Turner et al. (2006, 2008, 2012) and Laurent and Fennel (2019) are calibrated on May nitrate or
nitrate+nitrite loads from the Mississippi–Atchafalaya River Basin, assuming that the predicted HA in summers are under fair
weather. It is expected that models excluding wind effects can hardly produce accurate forecasts during summers with strong
winds or storms. Wind mixing effects on HA are considered in reaeration by introducing a wind stress term in the mechanistic
model (Obenour et al., 2015), while in the Bayesian model by Scavia et al (2013), the wind effects are considered indirectly
via an estimation based on current velocity and the reaeration rate given different wind conditions (i.e., fair weather, strong
westerly winds, and storms). However, as shown in Figure 1a, $PEA_{wind}$, which can also be interpreted as wind power, is found
poorly correlated to daily HA (R=-0.2458) compared to other highly correlated predictors and is dropped out of the candidate
list by the best subset searching approach. Forrest et al., (2011) also found that monthly wind power is not significantly
correlated to summer HA due to the short timescales of strong wind events. Therefore, the wind mixing effects considered by
Obenour et al (2015) and Scavia et al (2013) have limited contribution to the prediction of the interannual variability of the
HA. Indeed, our model construction process indicates that wind mixing, freshwater plume, and water temperature jointly
control the water stratification and vertical mixing, which directly modulates the reoxygenation of shelf water. PEA can serve
better in representing such effects rather than by wind speed or wind power alone. The daily PEA is significantly correlated to
daily HA (R=0.8178, $p<0.01$; Figure 1a) while the nonlinear effects of PEA cannot be neglected (Figure 5a). Therefore, an
accurate forecast of shelf hydrodynamics is critical for a robust summer HA prediction.

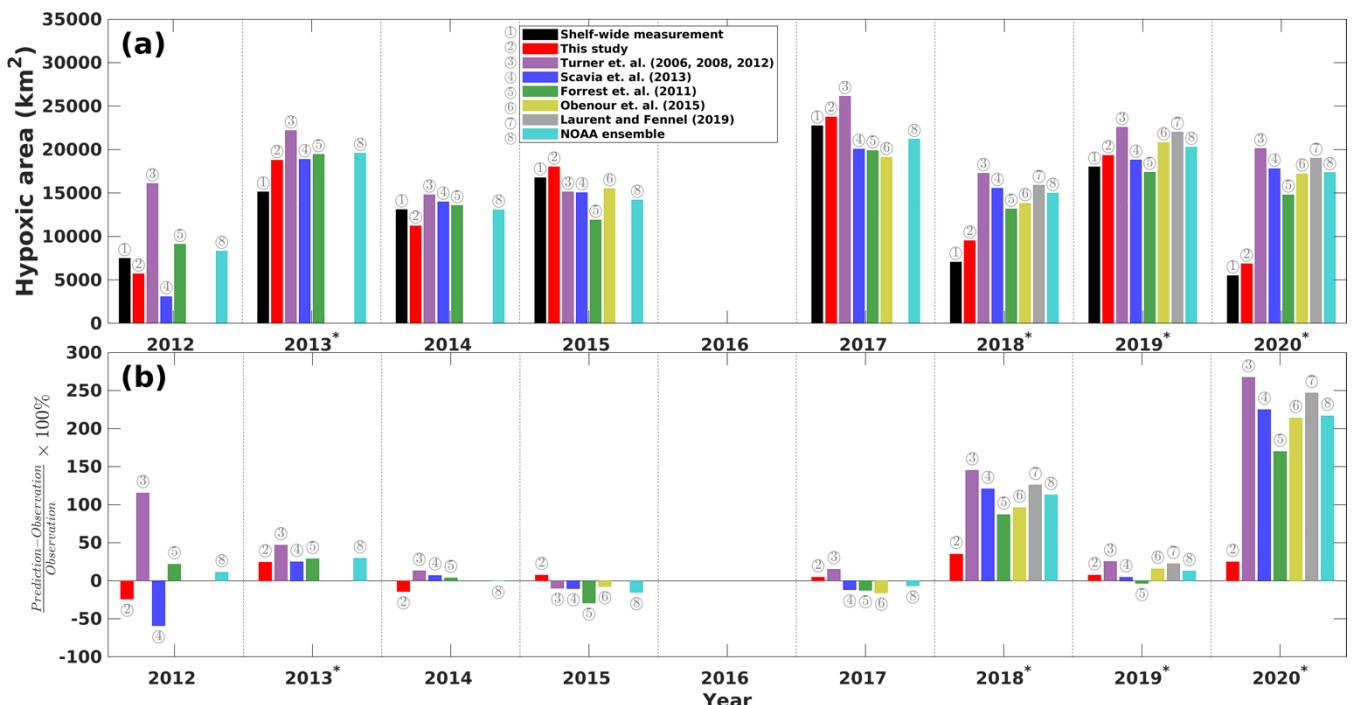


**Figure 8. (a) Comparisons of Shelf-wide measured and the best estimates of model predicted HA during the Shelf-wide cruise**
**periods. (b) Percentage differences between different model predictions and Shelf-wide measurements. The superscript asterisks**
**indicate high-wind years prior to the cruises.**

**Table 4 Statistics comparisons between model predictions and the Shelf-wide measurements. The R²s for predictions by Obenour et**
**al. (2015) and Laurent and Fennel (2019) are not given since the numbers of available records are small (N=5 and 3, respectively).**
**Numbers in paratheses indicate the numbers of compared records. Underscript "fair" and "windy" indicate that averages of**
**corresponding statistics are conducted for fair-weather and windy summers, respectively.**

|  | This study | Turner et al. (2006, 2008, 2012) | Scavia et al. (2013) | Forrest et al. (2011) | Obenour et al. (2015) | Laurent and Fennel (2019) | NOAA ensemble |
|---|---|---|---|---|---|---|---|
| $R^2$ | 0.9200 | 0.3017 | 0.2577 | 0.4061 | – | – | 0.3566 |
|  | (N=8) | (N=8) | (N=8) | (N=8) | (N=5) | (N=3) | (N=8) |
| RMSE (km) | 2005 | 7750 | 5797 | 4710 | 6412 | 9614 | 5460 |
|  | (N=8) | (N=8) | (N=8) | (N=8) | (N=5) | (N=3) | (N=8) |
| SI | 15 % | 59 % | 44 % | 36 % | 46 % | 95 % | 41 % |
|  | (N=8) | (N=8) | (N=8) | (N=8) | (N=5) | (N=3) | (N=8) |
| MAPB | 18 % | 80 % | 58 % | 44 % | 70 % | 132 % | 51 % |
|  | (N=8) | (N=8) | (N=8) | (N=8) | (N=5) | (N=3) | (N=8) |

| | | | | | | | |
|---|---|---|---|---|---|---|---|
| MAPB$_{fair-weather}$ | 15 % (N=4) | 46 % (N=4) | 25 % (N=4) | 18 % (N=4) | 8 % (N=2) | – (N=0) | 9 % (N=4) |
| MAPB$_{windy}$ | 18 % (N=4) | 58 % (N=4) | 40 % (N=4) | 33 % (N=4) | 43 % (N=3) | 74 % (N=3) | 40 % (N=4) |
| Data source (access in June 2022) | https://gulfhypoxia.net/ (Turner et al., 2006; 2008; 2012) <br> http://scavia.seas.umich.edu/hypoxia-forecasts/ (Scavia et al., 2013) <br> https://www.vims.edu/research/topics/dead_zones/forecasts/gom/index.php (Forrest et al., 2011) <br> https://obenour.wordpress.ncsu.edu/news/ (Obenour et al., 2015) <br> https://memg.ocean.dal.ca/news/ (Laurent and Fennel, 2019), <br> https://www.noaa.gov/news (NOAA ensemble) | | | | | | |


## 4.2 **Task force nutrient reduction**

In this section we assess the effects of nutrient reductions on HA using our model. Since 2001, the Mississippi River/Gulf of Mexico Hypoxia Task Force has set up a goal of controlling the size of mid-summer hypoxic zone below 5,000 km$^2$ in a 5-year running average (Mississippi River/Gulf of Mexico Watershed Nutrient Task Force, 2001; 2008) by reducing riverine nutrient loads. Because the monthly riverine silicate, phosphate, and nitrate+nitrite loads are highly correlated (Table A1), here we refer to nitrogen load (the only nutrient that has daily measurements) as the proxy for all riverine nutrients. The averaged summer HA during the Shelf-wide cruises in the most recent five years (2015, 2107, 2018, 2019, and 2020) are calculated with different nutrient reduction scenarios and are shown in Figure 9. The PEA, bottom temperature, and river discharges are unchanged, while the SOCalt is altered by reducing the nutrient concentration from 5% to 90%. The averaged observed HA is 14,000 km$^2$, while the averaged prediction by our ensemble model is 15,478 km$^2$, which is 11 % greater than the observation. As a leading time of 44 days (Figure A3a) is prescribed in SOCalt prior to Shelf-wide summer cruises in mid–late July, reduction strategies are applied to mid-June nutrient loads rather than May loads in our model. The monthly averaged total nitrogen loads for the 1980–1996 summers (April, May, and June) are $1.96 \times 10^8$ kg/month (Battaglin et al., 2010). It is comparable to the June-mean total nitrogen load ($1.6 \times 10^8$ kg month$^{-1}$) for the 2015–2020 period. We find that a 92 % reduction, which corresponds to a total nitrogen load of $5.5 \times 10^5$ kg day$^{-1}$ or $1.6 \times 10^7$ kg month$^{-1}$, is needed for the mid-June nutrient loads to achieve the goal of a 5,000 km$^2$ HA.

The recommended reduction strategy by our model is much more demanding than that by other models (Scavia et al., 2013; Obenour et al., 2015; Turner et al., 2012; Laurent and Fennel, 2019), which recommend a load reduction of 52 %–58 % related to the 1980–1996 average (Scavia et al., 2017). A recommendation of 92 % reduction is closed to that by Forrest et al. (2011) (80 %) when the coastal westerlies from 15 June to 15 July were considered in their regression model. Since water stratification is attributed to not only wind mixing effects but also effects from other physical processes (e.g., riverine freshwater transports

and surface heating), models developed based solely on May nutrient loads (Turner et al., 2012; Laurent and Fennel, 2019) or
nutrient loads and wind mixing (Scavia et al., 2013; Obenour et al., 2015) fail to capture water stratification's contribution to
hypoxia development. If a model considers the variability of HA to rely highly on the nutrient loads, then a moderate decrease
in nutrient loads would result in a substantial HA reduction. For further illustration, we re-ran the model without consideration
of the PEA (i.e., use $DCP_{Temp}$ and SOCalt or use only SOCalt). Model results show a substantial shrink of HA with moderately
reduced riverine nitrogen loads (Figure 9). In details, if only $DCP_{Temp}$ and SOCalt are used as the predictors, a nutrient reduction
by 60 % will satisfy the 5000 km$^2$ HA goal. And if we only use SOCalt as the predictor, then a 55 % in reduction is sufficient.
These results highlight the importance of considering PEA in HA predictions.

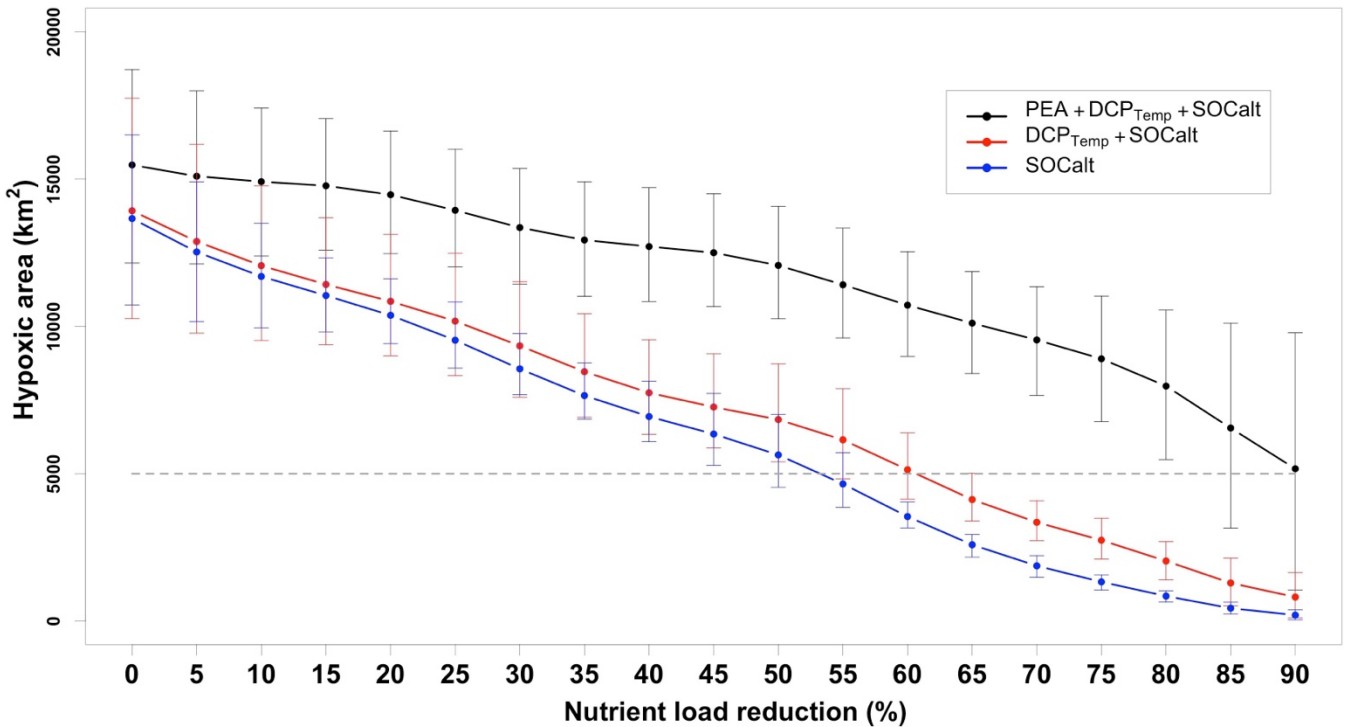


**Figure 9 2015–2020 mean (except 2016) of predicted HA in scenarios of different nutrient load reduction strategies given different**
**sets of predictors considered. Predictions by the ensemble model are conducted individually for the Shelf-wide cruise periods in**
**different summers and averaged from 2015 to 2020. Horizontal bars indicate ranges of 95 % PIs. Grey dashed lines represent the**
**goal of 5,000 km$^2$ set by the Mississippi River/Gulf of Mexico Hypoxia Task Force. Note here nutrient reduction percentages are**
**referred to mid-June nutrient loads in corresponding years.**
**5 Conclusion**
In this study, we present a novel HA forecast model for the LaTex Shelf using statistical analysis. The model is trained using
numeric simulations from 1 January 2007 to 26 August 2020 by a 3-dimensional coupled hydrodynamic–biogeochemical
model (ROMS). Multiple GLMs (regular Poisson GLMs, quasi-Poisson GLMs, negative binomial GLMs, zero-inflated
Poisson GLMs, and zero-inflated negative binomial GLMs) and GAMs (regular Poisson GAMs, quasi-Poisson GAMs, and

regular negative binomial GAMs) are assessed for HA predictions. Comparisons of model prediction performance illustrate that an ensemble model combing the prediction efforts of a zero-inflated Poisson GLM (GLMzip3) and a quasi-Poisson GAM (GAMqsp3) provides the most accurate HA forecast with PEA, SOCalt, and DCP$_{Temp}$ as predictors. The ensemble model is capable of explaining up to 77 % of the total variability of the hindcast HA and also provides a low RMSE of 3,256 km$^2$ and low MAPBs for overall (29 %) and peak predictions (25 %) when compared to the daily ROMS hindcasts.

We then applied the hydrodynamics field generated by a global model (HYCOM, GOFS 3.1) and performed a HA hindcast for the period from 1 January 2012 to 26 August 2020. The overall performance is barely acceptable with an R$^2$ of 0.4242, an RMSE of 5,088 km$^2$, and a SI of 38 % against the Shelf-wide summer cruise observations, largely due to HYCOM's relatively poor representation of shelf stratification. A substitution of ROMS-derived PEA led to a pronounced improvement with an R$^2$ of 0.9255, an RMSE of 3,751 km$^2$, and an SI of 28 %.

The ensemble model also provides an efficient yet more robust summer HA forecast than existing HA forecast models. Comparing against the Shelf-wide cruise observations, our model provides a high R$^2$ (0.9200 vs 0.2577–0.4061 by existing forecast models, same comparison hereinafter), a low RMSE (2,005 km$^2$ vs 4,710–9,614 km$^2$), a low SI (15 % vs 36 %–95 %), low MAPBs for overall (18 % vs 44 %–132 %), fair-weather summers (15 % vs 8 %–46 %), and windy summers (18 % vs 33 %–74 %) predictions. Sensitivity tests are conducted and suggests that a 92 % reduction in riverine nutrients related to the 1980–1996 summer average is required to meet the goal of a 5,000 km$^2$ HA. These results highlight the importance of considering PEA in HA prediction.

**Code/Data availability:** Model data is available at the LSU mass storage system and details are on the webpage of the Coupled Ocean Modeling Group at LSU (https://faculty.lsu.edu/zxue/). Data requests can be sent to the corresponding author via this webpage.

**Author contribution:** Bin Li and Z. George Xue designed the experiments and Yanda Ou carried them out. Yanda Ou developed the model code and performed the simulations. Yanda Ou, Bin Li, and Z. George Xue prepared the manuscript.

**Competing interests:** The authors declare that they have no conflict of interest.

**Acknowledgment:** Research support was provided through the Bureau of Ocean Energy Management (M17AC00019, M20AC10001). We thank Dr. Jerome Fiechter at UC Santa Cruz for sharing his NEMURO model codes. Computational support was provided by the High-Performance Computing Facility (clusters SuperMIC and QueenBee3) at Louisiana State University.


**Appendix A:**

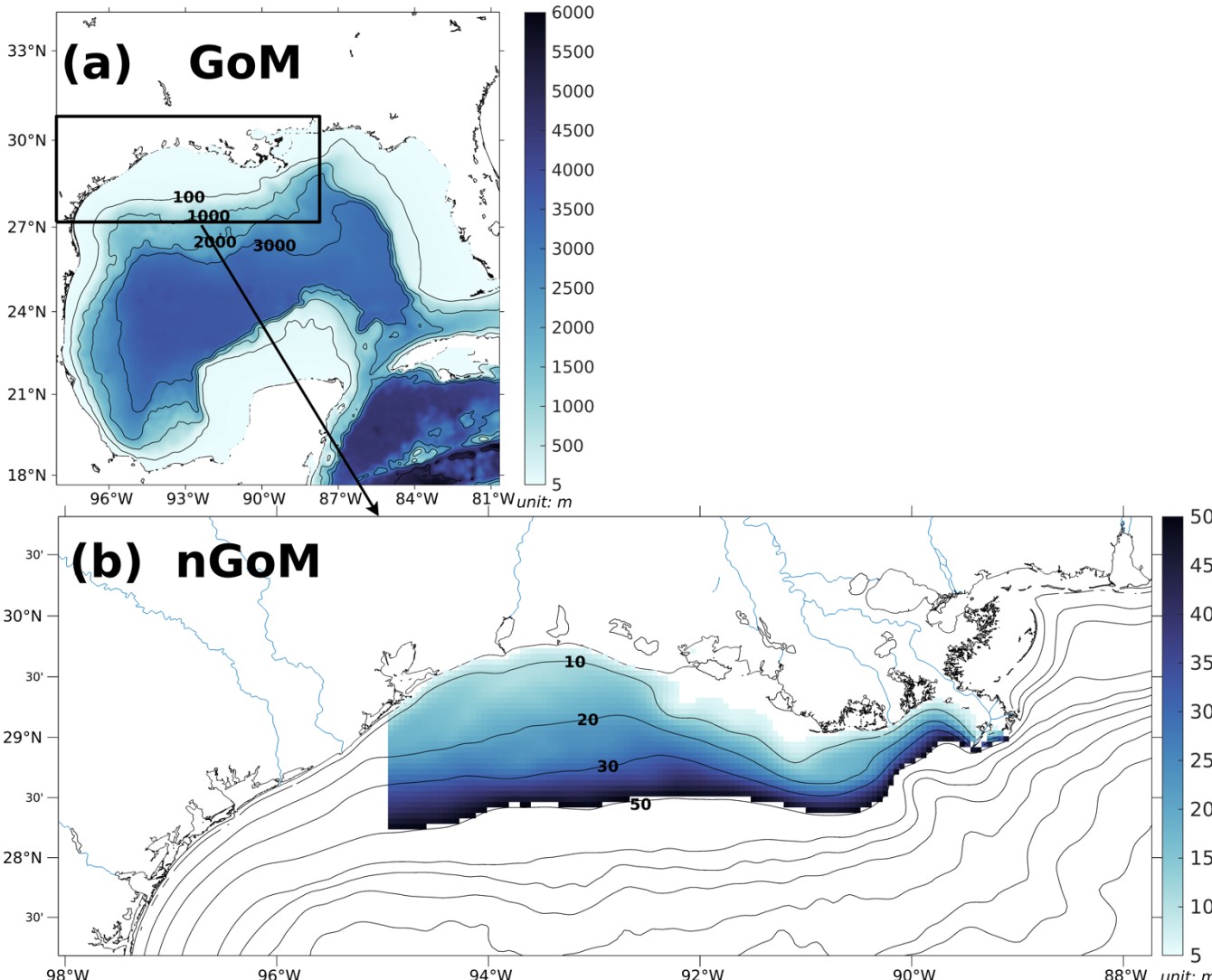

Figure A1 (a) Bathymetry of the entire domain of the Gulf–COAWST described in the accompanying study (Part I) and (b) zoom-
in bathymetry plot of the northern Gulf of Mexico (nGoM). The range of bathymetry of the color shaded area in (b) is from 6 to 50
m, over which the regional averages of parameters are conducted.

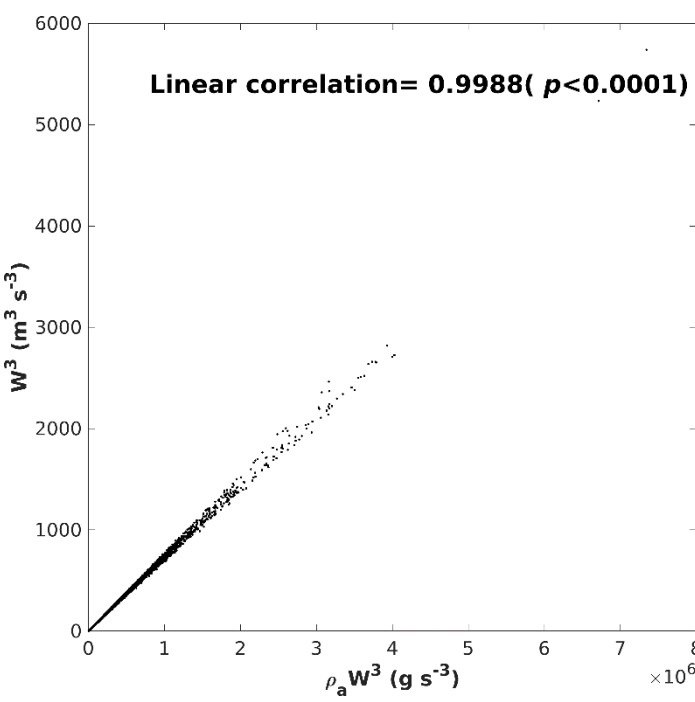


**Figure A2. A scatter plot of $\rho_a W^3$ against $W^3$ and their linear correlation.**
**Table A1 A correlation matrix of monthly mean inorganic nutrient loads by the Mississippi River and the Atchafalaya River from**
**2007 to 2020. Correlation coefficients shown are all significant ($p<0.001$).**

| | Mississippi nitrate+nitrite | Atchafalaya nitrate+nitrite | Mississippi phosphate | Atchafalaya phosphate | Mississippi silicate | Atchafalaya silicate |
|---|---|---|---|---|---|---|
| Mississippi nitrate+nitrite | 1 | | | | | |
| Atchafalaya nitrate+nitrite | 0.9207 | 1 | | | | |
| Mississippi phosphate | 0.8258 | 0.7551 | 1 | | | |
| Atchafalaya phosphate | 0.7576 | 0.7764 | 0.9308 | 1 | | |
| Mississippi silicate | 0.8511 | 0.7770 | 0.8664 | 0.7972 | 1 | |
| Atchafalaya silicate | 0.7989 | 0.7781 | 0.8147 | 0.7942 | 0.9673 | 1 |


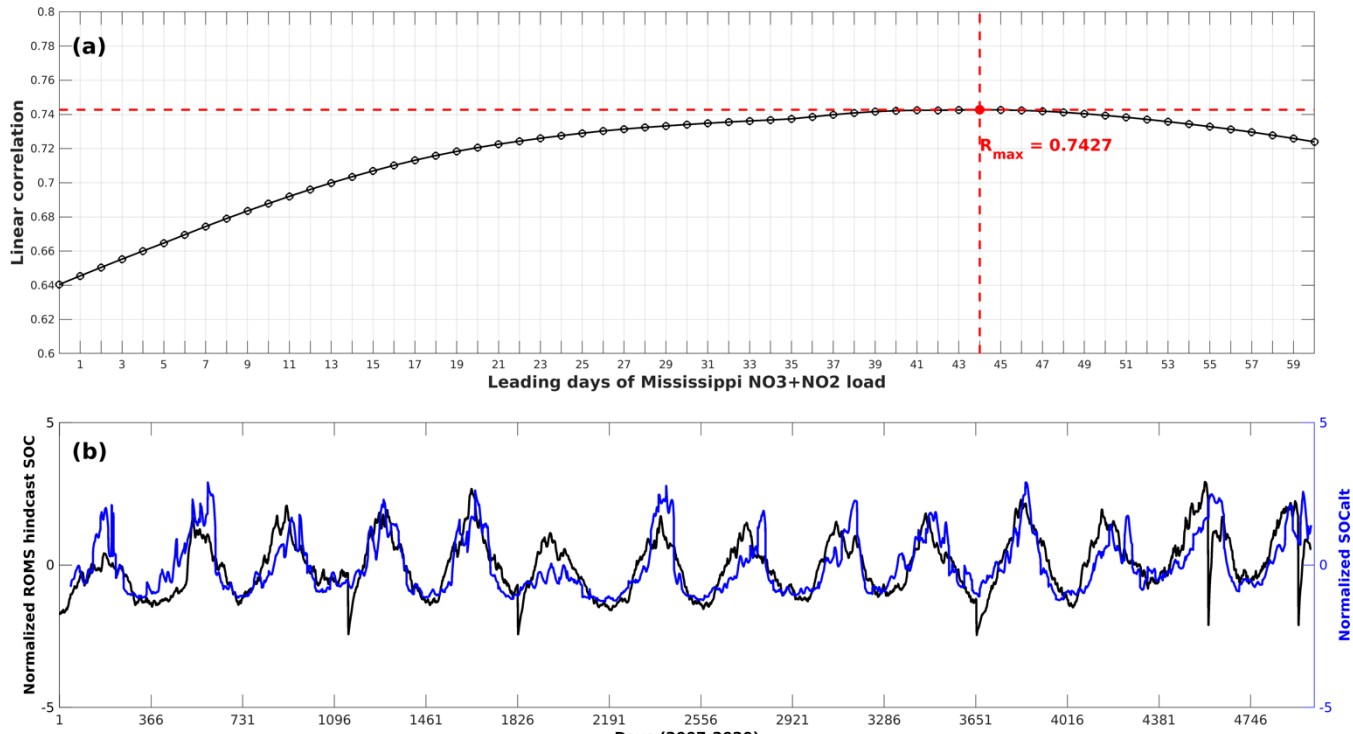

**Figure A3. (a) Lead/lag correlation coefficients between ROMS hindcast daily SOC and SOCalt ( = Mississippi River inorganic nitrogen loads · $e^{0.0693T_b}$) with the Mississippi nitrogen loads leading by different days; (b) daily time series of ROMS hindcast SOC and SOCalt when the Mississippi nitrogen loads leading by 44 days. The time series are regional average results over the LaTex Shelf and are normalized.**

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
