# Peer review of "Hydrodynamic and Biochemical Impacts on the Development of"

_Biogeosciences, 2022_

## Author Comment (AC1)

Responses to comments by Referee #1

Comment on bg-2022-4
Anonymous Referee #1

General Comments:
This manuscript applies an ensemble regression approach to produce daily predictions of hypoxic area for the Louisiana-Texas shelf. The manuscript is well written and provides more than adequate descriptions of the methods used to develop, train, and apply the multiple regression models considered for application. Although the ensemble model's application to global HYCOM seems an important aspect of this work, it's sole focus in the discussion feels somewhat like an afterthought. HYCOM presentation in the discussion also presents material that seems better suited for the methods section. Although not essential for publication (in my opinion), I would ask the authors to consider expanding the discussion of HYCOM application in the manuscript by addressing relevant technical details in the methods section and focusing on model outcomes for both the ROMS and HYCOM model in the discussion.

Authors' Responses: The main objective of this study is to apply the trained model to the HyCOM global dataset for an efficient forecast of hypoxic areas. As the application to the independent data (HyCOM) demonstrates the model's robustness, we treat it as a sole section and put it in the discussion section. Nevertheless, we would like to move specific technical details to the method section. We will expand the discussion section about the model's performance based on HyCOM to that found on ROMS and Shelfwide observations.

Specific Comments:

Line 55: "The effects of water column stratification are not included or only partially considered" In the previous paragraph you describe several models as incorporating water reaeration and wind velocity in the regression model. Are these not at minimum proxies for water column stratification? Suggest this be re-phrased to address the need to include stratification explicitly.

Authors' Responses: The previous models mentioned in the previous paragraph did consider stratification-relevant predictors like wind speed, water transport, and riverine nutrient loads (usually correlated to river discharges). However, water stratification is affected by all these variables and can be represented directly by the water density profiles. The statement of "the effects of water column stratification are not included or only partially considered…" may be improper and we could restate it as:

"The effects of water column stratification are considered only implicitly by the associated wind speeds, water transport, and riverine nutrient loads (usually highly correlated to river discharges), although stratification is documented as a crucial factor in regulating HA variability."

Line 155: where was the "temperature-dependent decomposition rate of organic matter" derived?

Authors' Responses: The SOC is modeled in the accompanying paper (Part I) (in Eq. (8) and Eq. (10)) proportional to sedimental organic matter concentration (estimated as sedimental particulate organic nitrogen, $PON_{sed}$, and is output from the 3-D coupled model) and a temperature-dependent decomposition rate:

$$SOC = PON_{sed} \cdot VP2N_0 \cdot e^{K_{P2N} \cdot T_b}$$

where $VP2N_0$ is a constant representing the decomposition rates of $PON_{sed}$ at 0 ºC, $K_{P2N}$ a constant (0.0693 ºC$^{-1}$) indicating temperature coefficients for decomposition of $PON_{sed}$, and $T_b$ the bottom water temperature. In this study, we use the variation of Mississippi River inorganic nitrogen loads with some leading days to represent the variation of $PON_{sed}$ and keep the temperature-dependent decomposition rate same as that of the 3-D coupled model. Such decomposition rate follows the Q$_{10}$ assumption that the reaction rate, R, depends exponentially on temperature, i.e.,

$$R = R_0 \cdot Q_{10}^{(T-T_0)/10}$$

For most biological systems, Q$_{10}$ is from 2 to 3. Here, we assume it as a constant 2. $R_0$ is the reaction rate at temperature $T_0$ (measured in ºC). The SOC scheme we applied takes the $R_0$ as $VP2N_0$ and $T_0$ as 0 ºC. Thus, the above equation can be simplified as:

$$R = R_0 \cdot 2^{\frac{T_b}{10}} \approx R_0 \cdot e^{0.0693 \cdot T_b}$$

Line 179: Is a shelfwide average appropriate for all predictors? Was any attempt made to derive predictors for different longitudinal zones of the shelf? For example, the ROMS grid extends far to the western limits of the hypoxic zone along Texas, and thus stratification predictors averaged across the entire domain may be less dynamic than otherwise expected.

Authors' Responses: We aim to develop a statistical model to predict the hypoxic area in the LaTex Shelf. Since the hypoxic area is the summation of the area which meets the hypoxic condition (Fig. 1 see below, we may need to add a map of LaTex Shelf in the Appendix), we averaged all the predictors over the same shelf region to represent the averaged hydrodynamic and biochemical conditions over this region. A median of the predictors over this region does also indicate averaged conditions and may provide a less PEA averaged condition (more dynamic) than using the regional average. However, models developed using regionally averaged predictors perform better than those built using median values.

Cruise surveys indeed observed hypoxic bottom water along the coastal Texas water (Fig .2). As one of the most well-known hypoxia cruises, the Shelfwide surveys do not usually cover waters west of 93.5W, however, in some summers (like 2017), the survey did reach the west of 94W along the Texas

coast where hypoxic water was reported. In addition, according to the SEAMAP summer Groundfish Survey (Fig. 2), hypoxic bottom water can be found to the west of 94W along the Texas coast. Thus, we expand the LaTex Shelf coverage which is larger than that shown in the Shelfwide cruises measurements. In this study, we have not yet considered the effects of location (like longitudes) on the hypoxic area prediction.

[Figure]

Fig. 1 Study domain.

Fig. 2 Distribution of bottom dissolved oxygen concentration provided by the SEAMAP Summer Groundfish Survey in 2011 summer.

Figure 4: The 95% CI are not visible. Is this because the confidence intervals are extremely tight and not visible at this multi-annual scale?
Authors' Responses: The ranges bagging 95% prediction intervals (PIs) are narrow and can be hardly seen here. We will choose another color to distinguish the PI and bagging mean curves.

Figure 7: Please clarify in the caption where the hypoxic area from the ROMS hindcast is coming from. Is it from the ensemble model, and if so, should 95% CI's be applied here?

Responses: The ROMS hindcast hypoxic area is from the 3-D coupled hydrodynamic-biogeochemical model described in the accompanying paper (PartI). For the application using HyCOM application, we trained the model using the ROMS hindcast results and then applied the trained model to the HyCOM dataset to predict the hypoxic area from 2019 to 2020. We then compared the predicted hypoxic area by the ensemble model using HyCOM and the ROMS hindcast hypoxic area in Figure 7.

Technical Corrections:

Authors' Responses: We will correct the below-mentioned sentences or phases according to the comments.

Abstract line 20: suggest changing "is by far the first one providing" to "is the first"

Line 24: superscript "L-1"

Line 56/57: "The information of future conditions is limited although some models are built upon multiple predictors, thus these forecast models are indeed "pseudo-forecast" ones." This sentence is awkward. Suggest rewording to: "Information on future conditions is often limited to few predictors, thus limiting these forecast models to "pseudo-forecasts""

Line145/146: "However, by far, global forecast model systems like HYCOM does not include biochemical fields" This sentence is a little confusing. When you say "fields", do you mean "parameters"? HyCOM is strictly a hydrodynamic model, so it is sufficient to say "However, global forecast models such as HYCOM do not simulate biochemical parameters. Therefore, the biochemical-related term SOC needs to be replaced by an alternative term (denoted as SOCalt)."

Line 150: It may be more appropriate to describe nitrogen as available for plankton growth, not bloom.

Line 164/165: "For simplification, we denoted this variable as (Qh), W3, and ð       as PEAheat, PEAwind, and DCPTemp, respectively." It took me a few reads to figure out what this sentence was trying to say. Suggest removing the word "this" and modifying to "denoted the variables"

Line 316: Change "It implies" to "This implies"

---

## Author Comment (AC2)

Responses to comments by Referee #2

Comment on bg-2022-4
Anonymous Referee #2

In their manuscript, Ou et al develop a novel statistical model to forecast/hindcast the size of the hypoxic area in the northern Gulf of Mexico. They use the model to test the feasibility of using HYCOM output and atmospheric data (reanalysis and forecast) to forecast the size of the hypoxic zone. The manuscript is well written and the statistical model seems to be able to retrieve the hypoxic area simulated with the ROMS model (part I paper). I am not familiar with the GLM/GAM statistical techniques and hopefully another reviewer can verify this part of the methodology. My overall assessment is that some improvements are required before the manuscript should be considered for publication. There are a few points that I think are important and would like raise below. Other, more specific comments are listed afterwards.

1) In its current form, the manuscript is mostly methodological and therefore I don't know if BG is the best fit for it. This could be solved with some improvements. For instance, the Discussion section presents an example of how to use the forecast model. This is a really interesting approach but it feels like a quick addition to justify the model development, that will be "further improved" in the future. A proper set of "forecasts" that are tested against observations would make a much more compelling case for the model's ability to forecast hypoxia. 1985-2021 mid-summer observations are available for this test; I believe that HYCOM and atmospheric forcing data are available in recent years to carry out this analysis. The forecast input data come with (high?) uncertainty and it would be interesting to know the effect on the hypoxia forecast (compared with the reanalysis input).

Authors' Response: We did compare the predicted hypoxic area by the ensemble model with the Shelfwide observations during the composition of this manuscript. We will expand our Discussion section by 1) comparing observed, predicted, ROM hindcast, and NOAA forecast hypoxic area; 2) assessing the sensitivity of input data to the predicted hypoxic area. We will also perform a long prediction according to HyCOM's availability.

2) My second point is a follow up from above. The manuscript relies exclusively on models. This is fine as a methodological paper but not if the authors aim at improving the current (seasonal) hypoxia forecasts and providing a tool for managers. For instance, it is assumed that the ROMS hindcast is a true representation of LaTex hypoxia. This is obviously not the case (as with any models) and it seems important to include observations in the manuscript to see how/if the forecasts drift away from the observations as we go from ROMS to GLM/GAM to HYCOM. Also note some of the reviewers comments on the Part I paper referenced here. Furthermore, the model provides a highly temporally resolved forecast, but it is not clear to me if, as a forecast, it does better than the seasonal forecast models (cited in the Introduction) that are, for some of them, spatially and temporally resolved. Some comparison with those (available annually through NOAA, e.g. https://www.noaa.gov/news-release/noaa-forecasts-averagesized-dead-zone-for-gulf-of-mexico) would strengthen the manuscript.

Authors' Response: We agree with this comment that observation should be involved in evaluating the model performance. The ROMS hindcast does capture the annual variability of the observed hypoxic area as in the Part I paper. It is also very interesting to compare our prediction with other forecast products like those by NOAA. We will add more discussion in the revision.
        To the best of our knowledge, no forecast model is capable of providing a hypoxia forecast map. We would like to resolve it in a future study.

3) The part that needs significant improvement is the Discussion, which is not really available in the current version of the manuscript. Rather, the Discussion section presents an attempt at a "real" forecast using HYCOM. This could be moved to the Results section and a real Discussion section should be provided. What does this new technique brings to the knowledge of LaTex hypoxia? How does it compare with earlier models? How is this useful to managers? What are the caveats and limitations? what are the future developments? How is this technique portable to other systems? All of those are legitimate points that should be discussed.

Authors' Response: We will improve the Discussion section by adding related discussions as recommended.

Specific comments

L36/53: Those are seasonal forecast and cannot include the wind since it is not predictable at this time scale.

Response: The model by Turner et al. (2006) was not built using wind information. Instead, it was built based on May nitrogen load and observations of the hypoxic area under fair weather. Therefore, their model can provide a robust annual prediction when no strong wind is present.

L56: Stratification is included indirectly in the statistical models

Response: The previous models as mentioned in the previous paragraph did consider stratification-relevant predictors like wind speed, water transport, and riverine nutrient loads (usually correlated to river discharges). However, water stratification is affected by all these variables and can be represented directly by the water density profiles. We plan to change this statement to "The effects of water column stratification are considered only implicitly by the associated wind speeds, water transport, and riverine nutrient loads (usually highly correlated to river discharges), although stratification is documented as a crucial factor in regulating HA variability."

L58: They are not pseudo forecast, they forecast the mid summer hypoxic area (well in advance). Therefore, they are seasonal forecasts, which is different from the short-term forecasts provided by HYCOM.

Response: We will use a seasonal forecast model instead of a pseudo forecast model here.

L58-59: "fail whenever winds are strong in summers": Note that some of these models provide information on the effect of the wind on the forecast

Response: The wind input they used is from historical records (e.g., Katin et al., 2021 and Laurent and Fennel, 2019). We will revise this sentence.

L76: FYI (related to the main comment above), looking at the comparison between ROMS and observed mid-summer hypoxic area in Part I manuscript, the r-square is 0.58.

L79: could you define the geographical limits that you use for the LaTex shelf? That would be helpful to have a sense of your comparisons as it is not clear if you use the same area as the mid-summer sampling cruises to calculate the hypoxic zone.

Response: The LaTex Shelf we mentioned here covers the region shown in the below figure (Fig. 1). We may need to add a map of the LaTex Shelf in the supplementary materials. The shelf region we chose is larger than the coverage of the Shelfwide cruise survey because 1) The Shelfwide cruise surveys do not usually extend to the west of 93.5W, however, in some summers (like 2017), the survey did reach the west of 94W along the Texas coastal where hypoxia was reported. 2) According to the SEAMAP summer Groundfish Survey (Fig. 2 below), hypoxic bottom water can be found to the west of 94W along with coastal Texas.

[Figure]

Fig. 1 Study domain.

Fig. 2 Distribution of bottom dissolved oxygen concentration provided by the SEAMAP Summer Groundfish Survey in 2011 summer.

L91: what do you mean by up to?
Response: We wanted to state that the correlation between PEA and SS is -0.88 which is high. We will correct this sentence as:
    Indeed, the correlation of regionally averaged PEA and SSS is significantly as high as -0.88 (p<0.001; Figure 1a) which emphasizes the importance of freshwater-induced stratification.

L148: It might be helpful to include these equations here.
Response: We will include these equations in the revision.

L158: Can you discuss the biological meaning of this time lag? It seems to indicate that mid-summer hypoxia is fuelled by early summer loads and therefore that there is no relationship between May load and summer hypoxia.
Response: The time lag here represents the time between the occurrence of massive organic matter consumption on the sediment and massive nutrient supply by rivers. The maximum nutrient loads do not always occur in May but sometimes in June and July (see below Fig. 3, we will provide daily time series

of riverine nutrients load in the supplementary materials). The correlation shown in Figure A2(a) is around 0.68 when the Mississippi nitrogen load leads by 60 days. The relationship between May load and mid-summer hypoxia is also statistically significant. But the correlation reaches the maximum when the Mississippi nitrogen load leads by 19 days.

[Figure]

Fig. 3 Daily time series of river discharge, nitrate supply, phosphate supply, and silicate supply by the Mississippi River and the Atchafalaya River from 2007 to 2020. Data is derived from the USGS.

L176 (Table 1): "Hypoxic area" would be better than "Area of extremely low dissolved oxygen concentration"
Response: We will correct it.

L194: Figures are not presented in order, please reorder
Response: We will move this sentence around Figure 4.

L198 (Figure 1a): The lack of relationship between SOCalt and botT is a bit concerning, can you comment?
Response: The SOCalt not only depends on $DCP_{Temp}$ but also on the riverine nutrient supply. Riverine nutrient supply does not reach the maximum as the temperature reaches the maximum in August. The maximum riverine nutrient supply was usually found from May to June. This is the reason why we had a low correlation between SOCalt and $DCP_{Temp}$.

L198 (Figure 1g): What is the time range of these data, all year, spring-summer, springfall?
Response: The date is provided daily from 1 January 2007 to 26 August 2020.

L223-228: I didn't get how this added term solves the high level of correlation between predictors
Response: The multicollinearity problem is hard to solve, but easy to be quantified by variance inflation factors (VIFs). We first developed the model and then quantified the multicollinearity among the selected predictors. As we stated in L259-260, the VIFs among the selected predictors are 2.60, 2.43, and 1.23 for

PEA, SOCalt, and DCPTemp, respectively. A VIFs value lower than 5 is usually considered as weak multicollinearity.

L258: "impaired"
Response: Corrected.

L264-274: Not sure if that is a good test of model skill. Excluding randomly half of the years (or 30-40%) would have provided a good dataset for testing. Can you discuss why you did not split the hypoxia data into years, since hypoxia is a seasonal process?
Response: We did not split the data by year but instead by the data feature of the hypoxic area. As shown in Figure 1(b), the distribution of daily hypoxic area is highly right-skewed with much fewer values in the medium- and high-value ranges. Splitting the data based on years does not guarantee the hypoxic area in the training set covers the entire range, which would weaken the model performance. Instead, we split the data maintaining the distribution of hypoxic area in both the training set and test set. More specifically, 80% of samples with hypoxic area within a given range (e.g., 0, (0, 5000], (5000, 10000], etc.) are chosen randomly for the training set, the rest 20% are put into the test set. Thus, the ranges of the hypoxic area in both sets can be guaranteed the same range as the entire dataset. This is the reason why we did not consider the daily data as a time series. Since hypoxia occurs annually, the low, medium and high range of hypoxic areas are corresponding to non-summer seasons, early summer, and mid- and late summer, although the comparison shown in Figure 4 looks like time series, it is not.

L281 (Figure 4): You should add observations.
L376: It is an interesting technique but lacks observations, why didn't you do a real forecast, i.e. a week ahead of the mid-summer cruise, for each year where the input data are available?
L373: Why not doing that for the entire time series?
L386: Your model forecast doesn't seem to do better than the seasonal forecast in 2019 and misses the pre-sampling mixing event, can you comment? The 2020 mid-summer hypoxic area is also largely overestimated (~20,000 vs 5,000) and seem to be doing worst than seasonal forecasts despite the model ability to take into account the effect of wind (there was a tropical storm before the mid summer sampling that year)
Response: We will compare our prediction with shelfwide observations, and the NOAA forecast for the entire time series.

The HyCOM global products can hardly capture all the hydrodynamical features due to the relatively low temporal resolution (monthly) of riverine forcing in the model. As shown in Figure 6a, the PEA is more underestimated in the HyCOM dataset than in the ROMS hindcast results. Even though the HyCOM products are scaled according to the relationship with the ROMS hindcast, the HyCOM-derived PEA is still overestimated or underestimated when compared to the ROMS hindcast. This is the main drawback of using the HYCOM products in forecasting. The low performance of the pre-sampling mixing event in 2019 can be attributed to this reason.

The 2020 mid-summer hypoxic area is not largely overestimated. The observed value is 5480 km$^2$ on around day 570 (Figure 7) when the prediction is around 7000 km$^2$.

L289: the correlation doesn't seem to be significant
Response: The SOCalt and DCP$_{Temp}$ are not significantly correlated. However, DCP$_{Temp}$ is a proxy of the decomposition rate of organic matter and also a proxy of sediment oxygen consumption. We cannot see the mechanism from the statistical regression model, but can attribute the significant coefficients to some explainable mechanisms according to the reference of predictors.

L293 (Table 2): What is Pr? does it make any sense to provide a Pr of <1e-16?
Response: Pr is the p-value. The p-values are all < 2E-16 not < 1E-16.

L299: "procedure"
Response: Corrected.

L316-317: Early summer or spring? It looks like hypoxia develops in Spring in the time series
Response: The comparison shown in Figure 4 is not a time series comparison. According to Figure 7 in the Part I paper, the hypoxia develops rapidly in late spring and early summer.

L316-323: Do you see all that in Figure 5?
Response: The predicted hypoxic area is a summation of s(PEA), s(SOCalt), and s($DCP_{Temp}$). Thus, a greater smooth function of the corresponding predictor indicates greater influences on the hypoxic area.

L343: This is not a discussion, see main comment above.
Response: We will improve our Discussion section according to the comments.

L377: "slight": ~20+% difference
Response: We will provide the percentage changes of prediction compared to the hindcast results to further illustrate it.

---

## Author Comment (AC3)

Comment on bg-2022-4
Anonymous Referee #3

The manuscript employs a novel approach in applying numerical and statistical modeling techniques to more accurately forecast hypoxia area on the Louisiana-Texas shelf in the Gulf of Mexico. After selecting a set of predictors that are well correlated with hypoxic area in the Gulf, a long-term ROMS numerical simulation of this study area (2007-2020) is used to train an ensemble of statistical models using both generalized linear and generalized additive modeling techniques. The most promising techniques are then applied to global model outputs and USGS forcings to develop an accurate forecast over a later time period (2019-2020).

Overall, the manuscript describes a highly applicable and useful approach to rapidly forecast hypoxic conditions using a statistical ensemble. This approach appears to offer multiple benefits to past forecasts and would serve as a helpful template for other coastal areas as well. The paper utilizes a limited number of explanatory variables to achieve a good fit, and I think that the predictors they use are appropriate and highly applicable to hypoxic area estimates. I've tried to include many notes to summarize these points, but this is not an exhaustive list.

**Major comments**

**General:**
There is a fair amount of general awkward phrasing and minor grammatical and spelling errors, but I don't find that they hinder my own understanding of the content.

**Introduction:**
I think that this section could be broken up into three sections as opposed to the 2 paragraphs it has now. Currently, only one sentence discusses the ecological/societal consequences of hypoxia in this region, and the authors immediately begin discussing the predictive capabilities of previous forecasting efforts. In my opinion, there could be more motivation in the first paragraph that illustrates why hypoxia forecasts are important and useful, and the benefits that environmental managers and others could gain from an accurate forecast. Otherwise, this reads a bit more like an interesting scientific modeling exercise done for its own sake. The second paragraph could then focus on past efforts to create a forecasting system, while the final paragraph could talk about some of the shortcomings that this model ensemble will address.
Authors' Response: We agree with Reviewer#3 and will rearrange the Introduction following the comment.

**Methods:**
I have some minor questions about the equations described for the hydrodynamic-related predictors section, but I don't think that they are likely to alter the conclusions of the paper in a meaningful way.

**Discussion:**
Would suggest renaming this section as "results" since a discussion section is typically what is described in the conclusions section here.

Authors' Response: We will move the application of HyCOM dataset to the Result session and rewrite our Discussion section by discussing some questions following reviewer #2's comments, which include *"What does this new technique brings to the knowledge of LaTex hypoxia? How does it compare with earlier models? How is this useful to managers? What are the caveats and limitations? what are the future developments? How is this technique portable to other systems?"*

**Conclusions**:

I think that the paper would benefit from a more comprehensive conclusion that reiterated some of the broader implications and benefits that could come from this hybrid ensemble approach. The final two sentences are really just devoted to saying that this is the first of its kind, which again reinforces some of the issues I mention in the introduction related to this being a pure modeling exercise.

Authors' Response: We will provide more discussion of the implications of this study and emphasize it in the Conclusion section.

**Specific Comments**

Line 15: It may benefit the reader to include a percentage value in comparison to the low RMSE value of 3204 square kilometers, which may be quite large in other coastal systems.

Authors' Response: We will add a percentage difference to illustrate the model performance in addition to RMSE and $R^2$.

Line 20: Suggest removing the words "by far". Because this model is the first to do this, the modifier "by far" suggests that no other groups are anywhere near this operational capability. I'm not sure if this is the intent, maybe this is meant instead to say that this ensemble model has the highest performance skill "by far".

Authors' Response: We intend to say that this ensemble model has the highest performance skill "by far". We will rewrite this sentence to avoid ambiguity, like by removing the phrase "by far".

Line 25: Suggest changing to "shelf-wide" here and elsewhere in the paper

Authors' Response: We will correct it accordingly.

Line 30: I've seen "destruction" of hypoxia used more often than "deconstruction" in the literature, suggest making this change

Authors' Response: We will correct it accordingly.

Line 41-43: Awkward phrasing, cut out "however" from sentence

Authors' Response: We will correct it accordingly.

Line 46-47: Suggest rephrasing as "An additional Bayesian model applied to summer bottom DO predictions accounts for May total nitrogen…"

Authors' Response: We will correct it accordingly.

Line 49-52: Suggest rewording as "Mechanistic prediction methods have also been applied by Laurent and Fennel (2019) to develop a weighted mean forecast that is calibrated using May nitrate loads and three-dimensional hindcast simulations over the period 1985-2018. Once calibrated, the model only requires May nitrate loads as an input to produce the seasonal forecast for a given year."

Authors' Response: We will correct it accordingly.

Line 55: Suggest changing "shortages" to "drawbacks"

Authors' Response: We will correct it accordingly.

Line 55-59: Remove periods before points 2 and 3, otherwise you can remove the colon and break them all up into single sentences. Point 2 could also be reworded slightly, reads awkwardly now. Change "year-to-year" to "interannual"

Authors' Response: We will correct it accordingly.

Line 61-62: Suggest rewording to something like "Here we aimed to provide a new

technique in HA prediction that considers both stratification and biochemical effects, and accurately produces daily forecasts of HA based on selected predictors' own forecasts."
Authors' Response: We will correct it accordingly.

Line 65-67: Hypoxic volume really hasn't been mentioned up to this point in the manuscript, and here you say that it will be neglected because HA is a better predictor anyway. Would suggest removing these sentences altogether.
Authors' Response: We will remove these sentences accordingly.

Line 71-77: I understand that some of the data used for model evaluation are described in the companion paper, but this section seems to be much more focused on derived model inputs (e.g. reanalyses and model outputs). Suggest changing the title of this section to reflect this better.
Authors' Response: We will change the title of this section to something like "Data preparation".

Line 87: Suggest changing to "… the amount of energy per volume required to homogenize the entire water column"
Authors' Response: We will correct the sentence accordingly.

Line 95: Change "… are other two factors influencing" to "are two other factors that influence"
Authors' Response: We will correct the sentence accordingly.

Line 95-96: Could be worth mentioning that the effect of tidal mixing on stratification is neglected in this study site, since it's included as an additional term in the Simpson 1981 paper.

Authors' Response: Yes indeed. The Simpson 1981 paper did consider a tidal mixing term which is ignored in our study. We will add a sentence "The effects of tidal mixing which was considered in Simpson's (1981) equation was neglected in our study due to the relatively weaker tidal effects on stratification in the shelf when compared to the effects of river and wind".

Line 98: The first term on the right-hand side of this equation is negative in Simpson et al. (1978), but it seems like the way that this has been defined (reversing the position of water density and depth-integrated water density), that this may actually be referencing the equation of Simpson 1981. Equation 1 in Simpson 1981 also does not have "h" in the first right-hand side term, but I'm unsure if this is an error on Simpson's part since it appears in the 1978 paper. Suggest changing the reference and/or modifying the equation (may be easier just to change the reference rather than redo calculations/figures).

Authors' Response: The "h" term in the first right-hand-side term should not be there if following Simpson's (1981) work. The potential energy anomaly in Simpson (1981) is a depth-averaged term while that in the Simpson et al. (1978) paper is not. We follow the potential energy anomaly equation in Simpson (1981). We will remove the "h" in our equation and redo our calculations and figures. We think the results would not be significantly changed due to the depth range in the shelf region is not quite large. The correlation of Q and Q*h is high as 0.99961 as shown below.

[Figure]

Line 110-111: Suggest referencing figure 1a here as was done in lines 90-92.
Authors' Response: We will add the referencing figure 1a here.

Line 126: Suggest changing "… estimated for the following" to "estimated by"
Authors' Response: We will correct this sentence accordingly.

Line 128: I am having trouble understanding why this equation does not match what is shown in equation 2.27 of Monteith and Unsworth (2014). It looks as if some simplification occurred such that the denominator of the exponential (T-T', where T'=36K in Monteith and Unsworth) was incorporated into the numerator in the manuscript. However, when I plot the two curves against each other I find that they are unequal, and the gap increases with increasing temperatures. At 20 degrees C, for example, this is equal to vapor pressure difference of approximately 23 Pa. Is this a relatively minor difference, or is this likely to strongly affect the correlation found when combined with W^3?

Authors' Response: Thanks for pointing it out. We did miss a T' in the numerator of the exponential term. We will correct the equation in the revision. We double-check the relationship of $W^3$ and the corrected $\rho_a$ and found the strong linear relationship still holds. The corrected figure (Figure A1) is shown below.

[Figure]

Line 142-143: Here I would also suggest pointing the reader to figure 1a as was done in lines 90-92.

Authors' Response: Figure reference will be added here.

Line 145-146: Suggest changing phrasing to "However, global forecast model systems like HYCOM do not currently include biochemical fields."

Authors' Response: We will rewrite this sentence accordingly.

Line 156: Suggest removing this sentence and adding the correlation metric to the sentence that describes it first from lines 153-155. This earlier sentence could then read "… calculated as 19 days ($R^2$=0.8157, Figure A2a)."

Authors' Response: We will adjust these sentences accordingly.

Line 158: Is there a reference for this decomposition rate coefficient, or has this described in more detail in the companion manuscript?

Authors' Responses: The SOC is modeled in the accompany paper (in Eq. (8) and Eq. (10)) proportional to sedimental organic matter concentration (estimated as sedimental particulate organic nitrogen, $PON_{sed}$, and is output from the 3-D coupled model in the accompany paper) and a temperature-dependent decomposition rate:

$$SOC = PON_{sed} \cdot VP2N_0 \cdot e^{K_{P2N} \cdot T_b}$$

where $VP2N_0$ is a constant representing the decomposition rates of $PON_{sed}$ at 0 ºC, $K_{P2N}$ a constant (0.0693 ºC$^{-1}$) indicating temperature coefficients for decomposition of $PON_{sed}$, and $T_b$ the bottom water temperature. In this study, we use the variation of Mississippi River inorganic nitrogen loads with some leading days to mimic the variation of $PON_{sed}$ and keep the temperature-dependent decomposition rate same as that in the 3-D coupled model. Such decomposition rate follows the Q10 assumption (van't Hoff, 1898) that the reaction rate, R, depends exponentially on temperature, i.e.,

$$R = R_0 \cdot Q_{10}^{(T-T_0)/10}$$

For most biological systems, Q10 is from 2 to 3 (Bryan et al., 2008). Here, we assume it as a constant 2. $R_0$ is the reaction rate at temperature $T_0$ (measured in ºC). The SOC scheme we applied takes the $R_0$ as $VP2N_0$ and $T_0$ as 0 ºC. Thus, the above equation can be simplified as:

$$R = R_0 \cdot 2^{\frac{T_b}{10}} \approx R_0 \cdot e^{0.0693 \cdot T_b}$$

van't Hoff, J. H. (1898). Lectures in theoretical and physical chemistry: Part I: Chemical dynamics LK (p.256). London: Edward Arnold.
https://UOLibraries.on.worldcat.org/oclc/220605730

Reyes, B. A., Pendergast, J. S., & Yamazaki, S. (2008). Mammalian peripheral circadian oscillators are temperature compensated. Journal of biological rhythms, 23(1), 95–98. https://doi.org/10.1177/0748730407311855

Line 163-165: I would suggest immediately describing these variables as PEA_heat, PEA_wind, and DCP_temp, rather than defining them here again.

Authors' Response: These variables have been described as they are derived in sections 2.1.1 and section 2.1.2. We thus will rewrite these two sentences to avoid prolixity.

Line 166: Can you better define what it means when you state that "multicollinearity may become a problem"? Maybe adding a short technical detail on the ramifications of this would be helpful to the reader.

Authors' Response: The multicollinearity indicates correlation among independent variables. When the multicollinearity problem occurs (i.e., strong correlations among independent variables are found), the assumption of independent variables is weakened or even collapses. It would lead to unreasonable coefficients for some highly correlated "independent" variables even though the response is well fitted by regression models. Thus, we should avoid this problem or make it less apparent. We did a best-subset searching for predictors and finally found three predictors (PEA, SOCalt, and $DCP_{Temp}$) that provided the best performance when applied to the GLMs and GAMs. The variance inflation factors (VIFs) of these predictors are 2.60, 2.43, and 1.23, respectively, which is less than 5 suggesting the violation of multicollinearity is negligible.

We will add the above description in our manuscript for further illustration.

Line 169-170: Are all the grid cells the same size for this model domain? Is this described in more detail in the companion paper?
Authors' Response: The sizes of the grid cells are not the same but are nearly a constant of $25.56\pm0.17$ $km^2$ (mean$\pm$1std). The minimum and maximum sizes are $25.18$ $km^2$ and $25.96$ $km^2$, respectively.

Line 188: Change "rest" to "remaining"
Authors' Response: We will change it accordingly.

Line 190-191: Change "is chosen randomly" to "are chosen randomly" and "is grouped into" to "are grouped into"
Authors' Response: We will change it accordingly.

Line 192: Suggest changing to "split at intervals of 5000 km^2"
Authors' Response: We will change it accordingly.

Line 272: Some awkward phrasing "… which impose more threatens to the shelf ecosystem."
Authors' Response: We wanted to emphasize that it is more important to increase the model performance in the hypoxic area peak during which the shelf ecosystem would face more threats than during the mild hypoxic events. We will rewrite this sentence accordingly.

Line 299: Misspelling of "procedure"
Authors' Response: We will correct it.

Line 332-333: Suggest change to "… tends to underestimate HA peak estimates (like those seen at samples 310 and 920)"
Authors' Response: We will correct it accordingly.

Line 351-352: What daily data are referred to here, the outputs derived from HYCOM or the nitrate and nitrite loadings from USGS?
Authors' Response: The daily data here are the HyCOM data and USGS nitrate and nitrite loads. We will rewrite this sentence to avoid misunderstanding.

Line 378-381: These two sentences are a bit repetitive and could be combined. I'm also not entirely clear about whether HYCOM is expected to integrate USGS runoff in the future. Is the use of daily estimates part of long-term plans for HYCOM simulations?
Authors' Response: We will simplify these two sentences. We are not sure if HyCOM modeling groups have such a plan for their global products.

Line 399: Some awkward phrasing, suggest changing to "… HA forecast capable of explaining up to 80% of the total variability"
Authors' Response: We will correct it accordingly.

Line 404: "… on HYCOM,s"

Authors' Response: It is a typo and should be "HyCOM". We will correct it accordingly.

---

## Author Response (AR1)

**Responses to comments by Referee #1**

Comment on bg-2022-4
Anonymous Referee #1

We re-ran the 3-dimensional coupled hydrodynamic–biogeochemical model described in the accompanying paper (Part I) following the reviewers' comments therein. All statistics are updated and we find no major changes in our results in this paper. We also moved the section "Application to Global Forecast Products (HYCOM)" from the "Discussion" section to "Model construction and results" section and expanded the Discussion section by adding model comparisons to the existing seasonal forecast models and recommendation to task force nutrient reduction strategy.

General Comments:
This manuscript applies an ensemble regression approach to produce daily predictions of hypoxic area for the Louisiana-Texas shelf. The manuscript is well written and provides more than adequate descriptions of the methods used to develop, train, and apply the multiple regression models considered for application. Although the ensemble model's application to global HYCOM seems an important aspect of this work, it's sole focus in the discussion feels somewhat like an afterthought. HYCOM presentation in the discussion also presents material that seems better suited for the methods section. Although not essential for publication (in my opinion), I would ask the authors to consider expanding the discussion of HYCOM application in the manuscript by addressing relevant technical details in the methods section and focusing on model outcomes for both the ROMS and HYCOM model in the discussion.

Authors' Responses: We moved the section "Application to Global Forecast Products (HYCOM)" to the "Model construction and results" section. We also added comparisons of HA predictions by HYCOM products, predictions by ROMS hindcast, and Shelf-wide summer cruises measurements covering the period of 2012–2020. The HYCOM performs relatively poor regarding to the stratification in the LaTex shelf. Model performance can be improved drastically by replacing the HYCOM-derived PEA with the ROMS-derived PEA ( the other two predictors remain unchanged as the HYCOM-derived ones). This result emphasizes the need of a reliable forecast of shelf hydrodynamics for a robust HA prediction.

Specific Comments:

Line 55: "The effects of water column stratification are not included or only partially considered"
In the previous paragraph you describe several models as incorporating water reaeration and wind velocity in the regression model. Are these not at minimum proxies for water column stratification? Suggest this be re-phrased to address the need to include stratification explicitly.

Authors' Responses: The models mentioned in the previous paragraph did consider stratification-relevant predictors like wind speed, water transport, and riverine nutrient loads (usually correlated to river discharges). However, water stratification is affected by all these variables and can be represented directly by the water density profiles. The statement of "the effects of water column stratification are not included or only partially considered…" may be improper and we have restated it as:

"The effects of water column stratification are considered only implicitly by the associated wind speeds, water transport, and riverine nutrient loads (usually correlated to river discharges), although stratification is documented as a crucial factor in regulating HA variability."

Line 155: where was the "temperature-dependent decomposition rate of organic matter" derived?

Authors' Responses: The SOC is modeled in the accompanying paper (Part I) (in Eq. (8) and Eq. (10)) proportional to sedimental organic matter concentration (estimated as sedimental particulate organic nitrogen, $PON_{sed}$, and is output from the 3-D coupled model) and a temperature-dependent decomposition rate. We have added the following equation into the updated manuscript as Eq. (9).

$$SOC = PON_{sed} \cdot VP2N_0 \cdot e^{K_{P2N} \cdot T_b}$$

where $VP2N_0$ is a constant representing the decomposition rates of $PON_{sed}$ at 0 °C, $K_{P2N}$ a constant (0.0693 °C$^{-1}$) indicating temperature coefficients for decomposition of $PON_{sed}$, and $T_b$ the bottom water temperature. In this study, we use the variation of Mississippi River inorganic nitrogen loads with some leading days to represent the variation of $PON_{sed}$ and keep the temperature-dependent decomposition rate same as that of the 3-D coupled model. Such decomposition rate follows the $Q_{10}$ assumption that the reaction rate, R, depends exponentially on temperature, i.e.,

$$R = R_0 \cdot Q_{10}^{(T-T_0)/10}$$

For most biological systems, $Q_{10}$ is from 2 to 3. Here, we assume it as a constant 2. $R_0$ is the reaction rate at temperature $T_0$ (measured in °C). The SOC scheme we applied takes the $R_0$ as $VP2N_0$ and $T_0$ as 0 °C. Thus, the above equation can be simplified as:

$$R = R_0 \cdot 2^{\frac{T_b}{10}} \approx R_0 \cdot e^{0.0693 \cdot T_b}$$

Line 179: Is a shelfwide average appropriate for all predictors? Was any attempt made to derive predictors for different longitudinal zones of the shelf? For example, the ROMS grid extends far to the western limits of the hypoxic zone along Texas, and thus stratification predictors averaged across the entire domain may be less dynamic than otherwise expected.

Authors' Responses: We aim to develop a statistical model to predict the hypoxic area in the LaTex Shelf. Since the hypoxic area is the summation of the area which meets the hypoxic condition, we averaged all the predictors over the same shelf region to represent the averaged hydrodynamic and

biochemical conditions over this region (color shaded area in Fig. 1b shown below, which has also been added to the updated manuscript in the Appendix A as Figure A1b). A median of the predictors over this region does also indicate averaged conditions and may provide less stratification (smaller PEA) than using the regional average. However, models developed using regionally averaged predictors perform better than those built using median values.

Cruise surveys indeed observed hypoxic bottom water along the coastal Texas water (Fig. 2 shown below). As one of the most well-known hypoxia cruises, the Shelf-wide surveys do not usually cover waters west of 93.5W, however, in some summers (like 2017), the survey did reach the west of 94W along the Texas coast where hypoxic water was reported. In addition, according to the SEAMAP summer Groundfish Survey (Fig. 2), hypoxic bottom water can be found to the west of 94W along the Texas coast. Thus, we expand the LaTex Shelf coverage which is larger than that shown in the Shelf-wide cruises measurements. In this study, we have not yet considered the effects of location (like longitudes) on the hypoxic area prediction.

[Figure]

**Fig. 1 (a) Bathymetry of the entire domain of the Gulf–COAWST described in the accompanying study (Part I) and (b) zoom-in bathymetry plot of the northern Gulf of Mexico (nGoM). The range of bathymetry of the color shaded area in (b) is from 6 to 50 m, over which the regional averages of parameters are conducted.**

[Figure]

**Fig. 2 Distribution of bottom dissolved oxygen concentration provided by the SEAMAP Summer Groundfish Survey in 2011 summer (source: SEAMAP).**

Figure 4: The 95% CI are not visible. Is this because the confidence intervals are extremely tight and not visible at this multi-annual scale?

Authors' Responses: The ranges bagging 95% prediction intervals (PIs) are narrow and can be hardly seen from this figure. We have replotted this figure (shown below as Fig. 3 and updated in the manuscript as Figure 4) with the green patches representing the 95% PIs.

[Figure]

**Fig. 3 Comparisons of model predicted HA and ROMS-hindcast HA in the test set. RMSEs and R²s are derived between model Bagging mean and ROMS-hindcast HA.**

Figure 7: Please clarify in the caption where the hypoxic area from the ROMS hindcast is coming from. Is it from the ensemble model, and if so, should 95% CI's be applied here?

Responses: In the previous submission, the ROMS hindcast hypoxic area is from the 3-D coupled hydrodynamic-biogeochemical model described in the accompanying paper (Part I). In this revision we updated Figure 7 in the manuscript (also shown as Fig. 4 below) showing a comparison of the Shelf-wide summer cruise observations against the prediction using our ensemble model based on pure HYCOM-derived predictors, the prediction based on hybrid HYCOM- and ROMS-derived (PEA) predictors. The comparisons highlight the important effects of water stratification on the HA predictions and also emphasize that accurate forecasts of shelf hydrodynamics are critical to a robust HA prediction model.

[Figure]

**Fig. 4 Comparisons of daily predicted HA by ensemble model ((GLMzip3+GAMqsp3)/2) when applied to adjusted HYCOM products and Shelf-wide measurements from 2012 to 2020. Model results shown in (a) are predicted using pure HYCOM-derived products (i.e., PEA, SOCalt, and DCP$_{Temp}$), while those in (b) are predicted by ROMS-derived PEA, HYCOM-derived SOCalt, and HYCOM-derived DCP$_{Temp}$. Discontinuity of the predictions is due to the lack of riverine nitrate+nitrite records at site USGS 07374000 in the Mississippi River.**

Technical Corrections:

Abstract line 20: suggest changing "is by far the first one providing" to "is the first"
Responses: The corresponding sentence has been removed as we updated our results and conclusion.

Line 24: superscript "L-1"
Responses: The "-1" has been superscripted.

Line 56/57: "The information of future conditions is limited although some models are built upon multiple predictors, thus these forecast models are indeed "pseudo-forecast" ones." This sentence is awkward. Suggest rewording to: "Information on future conditions is often limited to few predictors, thus limiting these forecast models to "pseudo-forecasts""
Responses: The corresponding sentence has been rephrased to be:
        "Forecast of the predictors is usually limited, which restricts some of these seasonal models to pseudo ones."

Line145/146: "However, by far, global forecast model systems like HYCOM does not include biochemical fields" This sentence is a little confusing. When you say "fields", do you mean "parameters"? HyCOM is strictly a hydrodynamic model, so it is sufficient to say "However, global forecast models such as HYCOM do not simulate biochemical parameters. Therefore, the biochemical-related term SOC needs to be replaced by an alternative term (denoted as SOCalt)."
Responses: The corresponding sentences has been rephrased to be:
        "However, global forecast models such as HYCOM do not cover biochemical parameters. Therefore, the biochemical-related term SOC needs to be replaced by an alternative term (denoted as SOCalt)."

Line 150: It may be more appropriate to describe nitrogen as available for plankton growth, not bloom.
Responses: We provided more details of the reasons why we chose the total nitrate+nitrite here. Please see below.
        "The total nitrate+nitrite loads by the Mississippi River are used to represent the PONsed variability due to the long-term data supports. The daily Mississippi River discharges at site 07374000 are updated daily by the U.S. Geological Survey (USGS) National Water Information System (NWIS) since March 2004. The total nitrogen concentration at site 07374000 is provided and updated daily by USGS since November 2011. Prior to 2011, nitrogen loads (at site 07374000) are provided monthly by USGS and, in this study, are interpolated to daily intervals according to the corresponding monthly loads. Although phosphate and silicate are another two limitation nutrients in the shelf, daily measurement are still not available for the Mississippi River. Monthly total nitrate+nitrite loads, phosphate loads, and silicate loads by both the Mississippi River and the Atchafalaya River are significantly correlated (Table A1). Therefore, the total nitrate+nitrite loads applied here can be interpreted as total nutrient loads by both river systems."

**Table A1 A correlation matrix of monthly mean inorganic nutrient loads by the Mississippi River and the Atchafalaya River from 2007 to 2020. Correlation coefficients shown are all significant ($p<0.001$).**

| | Mississippi nitrate+nitrite | Atchafalaya nitrate+nitrite | Mississippi phosphate | Atchafalaya phosphate | Mississippi silicate | Atchafalaya silicate |
|---|---|---|---|---|---|---|
| Mississippi nitrate+nitrite | 1 | | | | | |
| Atchafalaya nitrate+nitrite | 0.9207 | 1 | | | | |
| Mississippi phosphate | 0.8258 | 0.7551 | 1 | | | |
| Atchafalaya phosphate | 0.7576 | 0.7764 | 0.9308 | 1 | | |
| Mississippi silicate | 0.8511 | 0.7770 | 0.8664 | 0.7972 | 1 | |
| Atchafalaya silicate | 0.7989 | 0.7781 | 0.8147 | 0.7942 | 0.9673 | 1 |

Line 164/165: "For simplification, we denoted this variable as (Qh), W3, and ð        as PEAheat, PEAwind, and DCPTemp, respectively." It took me a few reads to figure out what this sentence was trying to say. Suggest removing the word "this" and modifying to "denoted the variables"

Responses: We have removed this sentence and rephrased the previous sentence to avoid ambiguity and prolixity. Please see below.

"As listed in Table 1, six candidate predictors are considered in the statistical models including four stratification-related variables (PEA, SSS, $PEA_{heat}$, and $PEA_{wind}$) and two bottom biochemical variables (SOCalt and $DCP_{Temp}$)."

Line 316: Change "It implies" to "This implies"
Responses: The sentence has been rephrased accordingly.

**Responses to comments by Referee #2**

Comment on bg-2022-4
Anonymous Referee #2

We re-ran the 3-dimensional coupled hydrodynamic–biogeochemical model described in the accompanying paper (Part I) following the reviewers' comments therein. All statistics are updated and we find no major changes in our results in this paper. We also moved the section "Application to Global Forecast Products (HYCOM)" from the "Discussion" section to "Model construction and results" section and expanded the Discussion section by adding model comparisons to the existing seasonal forecast models and recommendation to task force nutrient reduction strategy.

In their manuscript, Ou et al develop a novel statistical model to forecast/hindcast the size of the hypoxic area in the northern Gulf of Mexico. They use the model to test the feasibility of using HYCOM output and atmospheric data (reanalysis and forecast) to forecast the size of the hypoxic zone. The manuscript is well written and the statistical model seems to be able to retrieve the hypoxic area simulated with the ROMS model (part I paper). I am not familiar with the GLM/GAM statistical techniques and hopefully another reviewer can verify this part of the methodology. My overall assessment is that some improvements are required before the manuscript should be considered for publication. There are a few points that I think are important and would like raise below. Other, more specific comments are listed afterwards.

1) In its current form, the manuscript is mostly methodological and therefore I don't know if BG is the best fit for it. This could be solved with some improvements. For instance, the Discussion section presents an example of how to use the forecast model. This is a really interesting approach but it feels like a quick addition to justify the model development, that will be "further improved" in the future. A proper set of "forecasts" that are tested against observations would make a much more compelling case for the model's ability to forecast hypoxia. 1985-2021 mid-summer observations are available for this test; I believe that HYCOM and atmospheric forcing data are available in recent years to carry out this analysis. The forecast input data come with (high?) uncertainty and it would be interesting to know the effect on the hypoxia forecast (compared with the reanalysis input).

Authors' Response: We moved the section "Application to Global Forecast Products (HYCOM)"" to the "Model construction and results" section. We also expanded the Discussion section by adding model comparisons to existing seasonal forecast models and recommends to nutrient reduction strategy.

In the updated section of "Application to Global Forecast Products (HYCOM)", we updated Figure 7 showing a comparison of the Shelf-wide summer cruise observations against the prediction using our ensemble model based on pure HYCOM-derived predictors, the prediction based on hybrid HYCOM- and ROMS-derived (PEA) predictors. The comparisons highlight the important effects of water stratification on the HA predictions and also emphasize that accurate forecasts of shelf hydrodynamics are critical to a robust HA prediction model.

[Figure]

**Figure 7 Comparisons of daily predicted HA by ensemble model ((GLMzip3+GAMqsp3)/2) when applied to adjusted HYCOM products and Shelf-wide measurements from 2012 to 2020. Model results shown in (a) are predicted using pure HYCOM-derived products (i.e., PEA, SOCalt, and $DCP_{Temp}$), while those in (b) are predicted by ROMS-derived PEA, HYCOM-derived SOCalt, and HYCOM-derived $DCP_{Temp}$. Discontinuity of the predictions is due to the lack of riverine nitrate+nitrite records at site USGS 07374000 in the Mississippi River.**

In the Discussion section, we provided comparisons of HA among the predictions by our ensemble model, existing forecast models, and the Shelf-wide cruise measurements. We demonstrated that our model outcompetes other forecast model (Figure 8) for the recent summer (2012–2020) with a high $R^2$ (Table 4, 0.9200 vs 0.2577–0.4061 by other forecast models, same comparison hereinafter), a low RMSE (2,005 $km^2$ vs 4,710–9,614 $km^2$), a low SI (15 % vs 36 %–95 %), low MAPBs for overall (18 % vs 44 %–132 %), fair-weather summers (15 % vs 8 %–46 %), and windy summers (18 % vs 33 %–74 %) predictions. We also provided our recommendation (Figure 9) on the riverine nutrient reduction strategy and suggested that a 92 % reduction in nutrients regarding the 1980–1996 average is needed to meet the goal of 5,000 $km^2$ HA.

A better performance on shelf hydrodynamics can be achieved by using daily river discharges as forcings in the HYCOM or by using regional models, like ROMS, which can be forced by atmospheric forecast, HYCOM-derived boundary conditions, and daily river discharges. Once a reliable hydrodynamics forecast is available , a promising HA forecast can be achieved using our ensemble model. In addition, effects of nutrient dispersion on the bottom hypoxia evolution in different shelf waters need to be considered in future model development.

[Figure]

**Figure 8. (a) Comparisons of Shelf-wide measured and the best estimates of model predicted HA during the Shelf-wide cruise periods. (b) Percentage differences between different model predictions and Shelf-wide measurements. The superscript asterisks indicate high-wind years prior to the cruises.**

Table 4 Statistics comparisons between model predictions and the Shelf-wide measurements. The $R^2$s for predictions by Obenour et al. (2015) and Laurent and Fennel (2019) are not given since the numbers of available records are small (N=5 and 3, respectively). Numbers in paratheses indicate the numbers of compared records. Underscript "fair" and "windy" indicate that averages of corresponding statistics are conducted for fair-weather and windy summers, respectively.

| | This study | Turner et al. (2006, 2008, 2012) | Scavia et al. (2013) | Forrest et al. (2011) | Obenour et al. (2015) | Laurent and Fennel (2019) | NOAA ensemble |
|---|---|---|---|---|---|---|---|
| $R^2$ | 0.9200 (N=8) | 0.3017 (N=8) | 0.2577 (N=8) | 0.4061 (N=8) | – (N=5) | – (N=3) | 0.3566 (N=8) |
| RMSE (km) | 2005 (N=8) | 7750 (N=8) | 5797 (N=8) | 4710 (N=8) | 6412 (N=5) | 9614 (N=3) | 5460 (N=8) |
| SI | 15 % (N=8) | 59 % (N=8) | 44 % (N=8) | 36 % (N=8) | 46 % (N=5) | 95 % (N=3) | 41 % (N=8) |
| MAPB | 18 % (N=8) | 80 % (N=8) | 58 % (N=8) | 44 % (N=8) | 70 % (N=5) | 132 % (N=3) | 51 % (N=8) |
| MAPB_fair-weather | 15 % (N=4) | 46 % (N=4) | 25 % (N=4) | 18 % (N=4) | 8 % (N=2) | – (N=0) | 9 % (N=4) |
| MAPB_windy | 18 % (N=4) | 58 % (N=4) | 40 % (N=4) | 33 % (N=4) | 43 % (N=3) | 74 % (N=3) | 40 % (N=4) |
| Data source (access in June 2022) | https://gulfhypoxia.net/ (Turner et al., 2006; 2008; 2012) http://scavia.seas.umich.edu/hypoxia-forecasts/ (Scavia et al., 2013) https://www.vims.edu/research/topics/dead_zones/forecasts/gom/index.php (Forrest et al., 2011) https://obenour.wordpress.ncsu.edu/news/ (Obenour et al., 2015) https://memg.ocean.dal.ca/news/ (Laurent and Fennel, 2019), https://www.noaa.gov/news (NOAA ensemble) | | | | | | |

[Figure]

**Figure 9** 2015–2020 mean (except 2016) of predicted HA in scenarios of different nutrient load reduction strategies given different sets of predictors considered. Predictions by the ensemble model are conducted individually for the Shelf-wide cruise periods in different summers and averaged from 2015 to 2020. Horizontal bars indicate ranges of 95 % PIs. Grey dashed lines represent the goal of 5,000 km$^2$ set by the Mississippi River/Gulf of Mexico Hypoxia Task Force. Note here nutrient reduction percentages are referred to mid-June nutrient loads in corresponding years.

2) My second point is a follow up from above. The manuscript relies exclusively on models. This is fine as a methodological paper but not if the authors aim at improving the current (seasonal) hypoxia forecasts and providing a tool for managers. For instance, it is assumed that the ROMS hindcast is a true representation of LaTex hypoxia. This is obviously not the case (as with any models) and it seems important to include observations in the manuscript to see how/if the forecasts drift away from the observations as we go from ROMS to GLM/GAM to HYCOM. Also note some of the reviewers comments on the Part I paper referenced here. Furthermore, the model provides a highly temporally resolved forecast, but it is not clear to me if, as a forecast, it does better than the seasonal forecast models (cited in the Introduction) that are, for some of them, spatially and temporally resolved. Some comparison with those (available annually through NOAA, e.g. https://www.noaa.gov/news-release/noaa-forecasts-averagesized-dead-zone-for-gulf-of-mexico) would strengthen the manuscript.

Authors' Response: We added comparisons with the existing forecast models and Shelf-wide measurements in the Discussion section. Please see the above responses for a brief update and the updated manuscript for the details. For the hypoxic forecast map, we would like to resolve it in a future study as that will require a prediction of 2D (hypoxia extension) instead of 1D (hypoxic area).

3) The part that needs significant improvement is the Discussion, which is not really available in the current version of the manuscript. Rather, the Discussion section presents an attempt at a "real" forecast using HYCOM. This could be moved to the Results section and a real Discussion section

should be provided. What does this new technique brings to the knowledge of LaTex hypoxia? How does it compare with earlier models? How is this useful to managers? What are the caveats and limitations? what are the future developments? How is this technique portable to other systems? All of those are legitimate points that should be discussed.

Authors' Response: We moved the section "Application to HYCOM Global Dataset" to the "Model construction and results" section. We also expanded the Discussion section by adding model comparisons to existing seasonal forecast models and recommends to nutrient reduction strategy.

Specific comments

L36/53: Those are seasonal forecast and cannot include the wind since it is not predictable at this time scale.

Response: The model by Turner et al. (2006) was not built using wind information. Instead, it was built based on May nitrogen load and observations of the hypoxic area under fair weather. Therefore, their model can provide a robust annual prediction when no strong wind is present in summer.

L56: Stratification is included indirectly in the statistical models

Response: The previous models as mentioned in the previous paragraph did consider stratification-relevant predictors like wind speed, water transport, and riverine nutrient loads (usually correlated to river discharges). However, water stratification is affected by all of these variables and can be represented directly by the water density profiles.

We rewrote this statement as followed.
    "The effects of water column stratification are considered only implicitly by the associated wind speeds, water transport, and riverine nutrient loads (usually correlated to river discharges), although stratification is documented as a crucial factor in regulating HA variability."

L58: They are not pseudo forecast, they forecast the mid summer hypoxic area (well in advance). Therefore, they are seasonal forecasts, which is different from the short-term forecasts provided by HYCOM.

Response: We used a seasonal forecast model instead of a pseudo forecast model here. The sentence has been rewritten as following.
    "Forecast of the predictors is usually limited, which restricts some of these seasonal models to pseudo ones.

L58-59: "fail whenever winds are strong in summers": Note that some of these models provide information on the effect of the wind on the forecast

Response: The wind input they used is from historical records (e.g., Katin et al., 2021 and Laurent and Fennel, 2019). We rewrote this sentence.
    "Most models are only capable of capturing interannual HA variability and are not reliable in summers when winds are strong."

L76: FYI (related to the main comment above), looking at the comparison between ROMS and observed mid-summer hypoxic area in Part I manuscript, the r-square is 0.58.

L79: could you define the geographical limits that you use for the LaTex shelf? That would be helpful to have a sense of your comparisons as it is not clear if you use the same area as the mid-summer sampling cruises to calculate the hypoxic zone.

Response: The LaTex Shelf we mentioned here covers the region shown in the below figure (color shaded area shown in the Appendix A as Figure A1b in the updated manuscript). The shelf region we chose is larger than the coverage of the Shelfwide cruise survey because 1) The Shelfwide cruise surveys do not usually extend to the west of 93.5W, however, in some summers (like 2017), the survey did reach the west of 94W along the Texas coastal where hypoxia was reported. 2)

According to the SEAMAP summer Groundfish Survey (Fig. 1 below but not shown in the updated manuscript), hypoxic bottom water can be found to the west of 94W along with coastal Texas.

[Figure]

**Figure A1 (a) Bathymetry of the entire domain of the Gulf–COAWST described in the accompanying study (Part I) and (b) zoom-in bathymetry plot of the northern Gulf of Mexico (nGoM). The range of bathymetry of the color shaded area in (b) is from 6 to 50 m, over which the regional averages of parameters are conducted.**

[Figure]

**Fig. 1 Distribution of bottom dissolved oxygen concentration provided by the SEAMAP Summer Groundfish Survey in 2011 summer (source: SEAMAP).**

L91: what do you mean by up to?
Response: We wanted to state that the correlation between PEA and SS is -0.87 which is high. We have corrected this sentence as:

"Indeed, the correlation of regionally averaged PEA and SSS is significantly high as -0.87 ($p<0.001$; Figure 1a) which emphasizes the importance of freshwater-induced stratification."

L148: It might be helpful to include these equations here.
Response: We have included these equations in the revision as Eq. (9)–(10). We excerpted the associated description below.

$$SOC = PON_{sed} \cdot VP2N_0 \cdot e^{K_{P2N} \cdot T_b}, \tag{9}$$
$$SOCalt = \text{Mississippi River inorganic nitrogen loads (led by 44 days)} \cdot e^{0.0693 T_b}, \tag{10}$$

$VP2N_0$ is a constant representing the decomposition rates of sedimentary particulate organic nitrogen, $PON_{sed}$, at 0 ºC. $K_{P2N}$ is a constant (0.0693 ºC⁻¹) indicating temperature coefficients for decomposition of $PON_{sed}$. $T_b$ is bottom water temperature (in ˚C). The Q10 (= 2 given the above chosen coefficients) assumption is applied to mimic the aerobic decomposition rate of $PON_{sed}$.

L158: Can you discuss the biological meaning of this time lag? It seems to indicate that mid-summer hypoxia is fuelled by early summer loads and therefore that there is no relationship between May load and summer hypoxia.
Response: The time lag here represents the time between the occurrence of massive organic matter consumption in the sediment and nutrient supply by rivers. The maximum nutrient loads do not always occur in May but sometimes in June and July (see below Fig. 2, not shown in the updated manuscript). The correlation shown in Figure A3(a) (updated in the manuscript) is around 0.72 when the Mississippi nitrogen load leads by 60 days. The relationship between May load and mid-summer hypoxia is also statistically significant. But the correlation reaches the maximum when the Mississippi nitrogen load leads by 44 days.

In the Figure A3(a), the leading days when the maximum correlation is found changed as we updated our ROMS hindcast with more reasonable setups (sinking velocity was changed from 15 m day$^{-1}$ to 5 m day$^{-1}$).

[Figure]

**Fig. 2 Daily time series of river discharge, nitrate supply, phosphate supply, and silicate supply by the Mississippi River and the Atchafalaya River from 2007 to 2020. Data is derived from the USGS.**

[Figure]

**Figure A3. (a) Lead/lag correlation coefficients between ROMS hindcast daily SOC and SOCalt ( = Mississippi River inorganic nitrogen loads · e$^{0.0693T_b}$ ) with the Mississippi nitrogen loads leading by**

different days; (b) daily time series of ROMS hindcast SOC and SOCalt when the Mississippi nitrogen loads leading by 44 days. The time series are regional average results over the LaTex Shelf and are normalized.

L176 (Table 1): "Hypoxic area" would be better than "Area of extremely low dissolved oxygen concentration"
Response: We have corrected it in Table 1.

L194: Figures are not presented in order, please reorder
Response: We removed the footnote for Figure 4 and restated these sentences in the second paragraph in section 3.2.2 where Figure 4 is firstly described. The sentences added in the section 3.2.2 are excerpted below.

"It is noted that the training set and test set are resampled according to different HA intervals. Since the distributions of HA in each year are similar (see Section 2.2), HA in both training and test set contains observations of peak and non-peak values in each year. Therefore, samples shown in Figure 4 are listed sequentially in the time dimension from 2007 to 2020 but are not necessarily evenly distributed."

L198 (Figure 1a): The lack of relationship between SOCalt and botT is a bit concerning, can you comment?
Response: The SOCalt not only depends on $DCP_{Temp}$ but also on the riverine nutrient supply. Riverine nutrient supply does not reach the maximum as the temperature reaches the maximum in August. The maximum riverine nutrient supply was usually found from May to June. This is the reason why we had a low correlation between SOCalt and $DCP_{Temp}$.

L198 (Figure 1g): What is the time range of these data, all year, spring-summer, springfall?
Response: The date is provided daily from 1 January 2007 to 26 August 2020. We have added it to the description of Figure 1.

L223-228: I didn't get how this added term solves the high level of correlation between predictors
Response: The multicollinearity problem is hard to solve, but can be quantified by variance inflation factors (VIFs). We first developed the model and then quantified the multicollinearity among the selected predictors. The VIFs among the selected predictors are found as 2.15, 2.70, and 1.59 for PEA, SOCalt, and DCPTemp, respectively. A VIFs value lower than 5 is usually considered as weak multicollinearity.

L258: "impaired"
Response: Corrected.

L264-274: Not sure if that is a good test of model skill. Excluding randomly half of the years (or 30-40%) would have provided a good dataset for testing. Can you discuss why you did not split the hypoxia data into years, since hypoxia is a seasonal process?
Response: We did not split the data by year but instead by the data feature of the hypoxic area. As shown in Figure 1(b), the distribution of daily hypoxic area is highly right-skewed with much fewer values in the medium- and high-value ranges. Splitting the data based on years does not guarantee the hypoxic area in the training set covers the entire range of hypoxia water extension, which would weaken the model performance. Instead, we split the data maintaining the distribution

of hypoxic area in both the training set and test set. More specifically, 80% of samples with hypoxic area within a given range (e.g., 0, (0, 5000], (5000, 10000], etc.) are chosen randomly for the training set, the rest 20% are put into the test set. Thus, the ranges of the hypoxic area in both test and training sets can be guaranteed the same range as the entire dataset. This is the reason why we did not consider the daily data as a time series. Since hypoxia occurs annually, the low, medium and high range of hypoxic areas are corresponding to non-summer seasons, early summer, and mid- and late summer, although the comparisons shown in Figure 4 looks like time series, it is not evenly distributed along time.

L281 (Figure 4): You should add observations.
L376: It is an interesting technique but lacks observations, why didn't you do a real forecast, i.e. a week ahead of the mid-summer cruise, for each year where the input data are available?
L373: Why not doing that for the entire time series?
L386: Your model forecast doesn't seem to do better than the seasonal forecast in 2019 and misses the pre-sampling mixing event, can you comment? The 2020 mid-summer hypoxic area is also largely overestimated (~20,000 vs 5,000) and seem to be doing worst than seasonal forecasts despite the model ability to take into account the effect of wind (there was a tropical storm before the mid summer sampling that year)

Response: We updated our Discussion section by adding model comparisons to the prevailing seasonal forecast models and recommends to nutrient reduction strategy. Our model results did produce better HA predictions for summers from 2012 to 2020 when compared to the Shelf-wide measurements than other seasonal forecast models.

The HYCOM global products can hardly capture the shelf stratification due to the relatively low temporal resolution (monthly) of riverine forcing. As shown in Figure 6a, the PEA is more underestimated in the HYCOM dataset than in the ROMS hindcast results. This is the main drawback of using the HYCOM products in HA forecast. The poor performance of the mixing event prior to the 2019 cruise can be attributed to this reason. The 2020 mid-summer hypoxic area is not largely overestimated. The observed value is 5480 km$^2$ while the prediction is 7788 km$^2$ when applied to the HYCOM-derived predictors (Figure 7a, updated and shown below).

[Figure]

**Figure 7. Comparisons of daily predicted HA by ensemble model ((GLMzip3+GAMqsp3)/2) when applied to adjusted HYCOM products and Shelf-wide measurements from 2012 to 2020. Model results shown in (a) are predicted using pure HYCOM-derived products (i.e., PEA, SOCalt, and DCP$_{Temp}$), while those in (b) are predicted by ROMS-derived PEA, HYCOM-derived SOCalt, and HYCOM-derived DCP$_{Temp}$. Discontinuity of**

**the predictions is due to the lack of riverine nitrate+nitrite records at site USGS 07374000 in the Mississippi River.**

L289: the correlation doesn't seem to be significant

Response: The SOCalt and DCP$_{Temp}$ are not significantly correlated. However, DCP$_{Temp}$ is a proxy of the decomposition rate of organic matter and also a proxy of sediment oxygen consumption. We cannot see the mechanism from the statistical regression model but can attribute the significant regression coefficients to some explainable mechanisms according to the reference of predictors.

L293 (Table 2): What is Pr? does it make any sense to provide a Pr of <1e-16?

Response: Pr is the p-value. The p-values are all < 2E-16 not < 1E-16.

L299: "procedure"

Response: Corrected.

L316-317: Early summer or spring? It looks like hypoxia develops in Spring in the time series

Response: The comparison shown in Figure 4 is not a time series comparison. According to Figure 7 in the Part I paper, the hypoxia develops rapidly in late spring and early summer.

L316-323: Do you see all that in Figure 5?

Response: The log of the predicted hypoxic area is a summation of s(PEA), s(SOCalt), and s(DCP$_{Temp}$). Thus, a greater smooth function of the corresponding predictor indicates greater influences on the hypoxic area.

L343: This is not a discussion, see main comment above.

Response: We have updated the Discussion section and provided a brief description in above responses.

L377: "slight": ~20+% difference

Response: The comparison between predictions using HYCOM products and hindcast by ROMS has been replaced by the comparison between predictions and Shelf-wide measurements. We introduced scatter index (SI, see below and also Eq. (15) in the updated manuscript) to illustrate the relative difference between predictions and observations (Figure 7). In addition, mean absolute percentage bias (MAPB, see below and also Eq. (14) in the updated manuscript) is also used as another statistics for relative difference between perditions and ROMS hindcast (see below and also the updated Figure 4 and added Table 2 in the newly revised manuscript).

$$MAPB = \frac{1}{N}\sum_{i=1}^{N}\left|\frac{P_i - O_i}{O_i}\right| \times 100\% \tag{14}$$

$$SI = \frac{RMSE}{\bar{O}} \times 100\% \tag{15}$$

[Figure]

**Figure 4. Comparisons of model predicted HA and ROMS-hindcast HA in the test set. RMSEs and R²s are derived between model Bagging mean and ROMS-hindcast HA.**

**Table 2 Mean absolute percentage bias between predicted and hindcast HA in the test set within different ranges of hindcast HA. The mean bias when hindcast HA < 5,000 km² is not shown since the prediction accuracy at high HA ranges is a more important feature of HA prediction models. The threshold of 5,000 km² is chosen because it is the goal HA set by the Action Plan (Mississippi River/Gulf of Mexico Watershed Nutrient Task Force, 2001; 2008). HA above this threshold is more worthy of attention.**

| Hindcast HA range (km²) | GLMzip3 | GAMqsp3 | Ensemble |
|---|---|---|---|
| [5000, 10000] | 38 | 40 | 36 |
| [10000, 20000] | 32 | 25 | 28 |
| [20000, 30000] | 34 | 26 | 28 |
| ≥ 30000 | 29 | 28 | 25 |
| Average | 33 | 30 | 29 |

**Responses to comments by Referee #3**

Comment on bg-2022-4
Anonymous Referee #3

General responses:

We re-ran the 3-dimensional coupled hydrodynamic–biogeochemical model described in the accompanying paper (Part I) following the reviewers' comments therein. All statistics are updated and we find no major changes in our results in this paper. We also moved the section "Application to Global Forecast Products (HYCOM)" from the "Discussion" section to "Model construction and results" section and expanded the Discussion section by adding model comparisons to the existing seasonal forecast models and recommendation to task force nutrient reduction strategy.

The manuscript employs a novel approach in applying numerical and statistical modeling techniques to more accurately forecast hypoxia area on the Louisiana-Texas shelf in the Gulf of Mexico. After selecting a set of predictors that are well correlated with hypoxic area in the Gulf, a long-term ROMS numerical simulation of this study area (2007-2020) is used to train an ensemble of statistical models using both generalized linear and generalized additive modeling techniques. The most promising techniques are then applied to global model outputs and USGS forcings to develop an accurate forecast over a later time period (2019-2020).

Overall, the manuscript describes a highly applicable and useful approach to rapidly forecast hypoxic conditions using a statistical ensemble. This approach appears to offer multiple benefits to past forecasts and would serve as a helpful template for other coastal areas as well. The paper utilizes a limited number of explanatory variables to achieve a good fit, and I think that the predictors they use are appropriate and highly applicable to hypoxic area estimates. I've tried to include many notes to summarize these points, but this is not an exhaustive list.

**Major comments**

**General:**
There is a fair amount of general awkward phrasing and minor grammatical and spelling errors, but I don't find that they hinder my own understanding of the content.

**Introduction:**
I think that this section could be broken up into three sections as opposed to the 2 paragraphs it has now. Currently, only one sentence discusses the ecological/societal consequences of hypoxia in this region, and the authors immediately begin discussing the predictive capabilities of previous forecasting efforts. In my opinion, there could be more motivation in the first paragraph that illustrates why hypoxia forecasts are important and useful, and the benefits that environmental managers and others could gain from an accurate forecast. Otherwise, this reads a bit more like an interesting scientific modeling exercise done for its own sake. The second paragraph could then focus on past efforts to create a forecasting system, while the final paragraph could talk about some of the shortcomings that this model ensemble will address.

Response: We have rearranged the Introduction following the comment.
In the first part, the influences of hypoxia on the fish displacements and the distribution of Gulf shrimp fleets are added. In the second part, a brief description of existing forecast models is presented. In the last part, we pointed out the disadvantage of existing forecast models and

addressed our aim of this study, i.e., to provide a new technique in HA prediction that considers both stratification and biochemical effects, and accurately produces daily forecasts of HA based on selected predictors' forecasts.

**Methods:**

I have some minor questions about the equations described for the hydrodynamic-related predictors section, but I don't think that they are likely to alter the conclusions of the paper in a meaningful way.

**Discussion:**

Would suggest renaming this section as "results" since a discussion section is typically what is described in the conclusions section here.

Authors' Response: We moved the section "Application to Global Forecast Products (HYCOM)" to the "Model construction and results" section. We also expanded the Discussion section by adding model comparisons to existing seasonal forecast models and recommends to nutrient reduction strategy.

In the Discussion section, we provided comparisons of HA among the predictions by our ensemble model, existing forecast models, and the Shelf-wide cruise measurements. We demonstrated that our model outcompetes other forecast model for the recent summer (2012–2020) with a high $R^2$ (0.9200 vs 0.2577–0.4061 by other forecast models, same comparison hereinafter), a low RMSE (2,005 $km^2$ vs 4,710–9,614 $km^2$), a low SI (15 % vs 36 %–95 %), low MAPBs for overall (18 % vs 44 %–132 %), fair-weather summers (15 % vs 8 %–46 %), and windy summers (18 % vs 33 %–74 %) predictions. We also provided our recommendation on the riverine nutrient reduction strategy and suggested that a 92 % reduction in nutrients regarding the 1980–1996 average is needed to meet the goal of 5,000 $km^2$ HA.

[Figure]

**Figure 7. Comparisons of daily predicted HA by ensemble model ((GLMzip3+GAMqsp3)/2) when applied to adjusted HYCOM products and Shelf-wide measurements from 2012 to 2020. Model results shown in (a) are predicted using pure HYCOM-derived products (i.e., PEA, SOCalt, and DCPTemp), while those in (b) are predicted by ROMS-derived PEA, HYCOM-derived SOCalt, and HYCOM-derived DCPTemp.**

**Discontinuity of the predictions is due to the lack of riverine nitrate+nitrite records at site USGS 07374000 in the Mississippi River.**

**Conclusions:**

I think that the paper would benefit from a more comprehensive conclusion that reiterated some of the broader implications and benefits that could come from this hybrid ensemble approach. The final two sentences are really just devoted to saying that this is the first of its kind, which again reinforces some of the issues I mention in the introduction related to this being a pure modeling exercise.

Response: We addressed that the implications of this study as followed.

1) In the Discussion section, we provided comparisons of HA among the predictions by our ensemble model, predictions by the existing NOAA-supported forecast models, and the Shelf-wide cruise measurements (added Figure 8 and Table 4). We demonstrated that our model outcompetes the prevailing forecast model for the recent summer (2012–2020) with a high $R^2$ (0.9200 vs 0.2577–0.4061 by existing forecast models, same comparison hereinafter), a low RMSE (2,005 $km^2$ vs 4,710–9,614 $km^2$), a low SI (15 % vs 36 %–95 %), low MAPBs for overall (18 % vs 44 %–132 %), fair-weather summers (15 % vs 8 %–46 %), and windy summers (18 % vs 33 %–74 %) predictions.

2) The recommendation of nutrient reduction strategy which is found more harsher that what the prevailing forecast models suggest. A 92 % reduction in nutrients related to the 1980–1996 average is needed to meet the hypoxia reduction goal of 5,000 $km^2$ (added Figure 9).

[Figure]

**Figure 8. (a) Comparisons of Shelf-wide measured and the best estimates of model predicted HA during the Shelf-wide cruise periods. (b) Percentage differences between different model predictions and Shelf-wide measurements. The superscript asterisks indicate high-wind years prior to the cruises.**

**Table 4 Statistics comparisons between model predictions and the Shelf-wide measurements. The $R^2$s for predictions by Obenour et al. (2015) and Laurent and Fennel (2019) are not given since the numbers of available records are small (N=5 and 3, respectively). Numbers in paratheses indicate the numbers of compared records. Underscript "fair" and "windy" indicate that averages of corresponding statistics are conducted for fair-weather and windy summers, respectively.**

|  | This study | Turner et al. (2006, 2008, | Scavia et al. (2013) | Forrest et al. (2011) | Obenour et al. (2015) | Laurent and Fennel | NOAA ensemble |
|---|---|---|---|---|---|---|---|

| | | 2012) | | | | (2019) | |
|---|---|---|---|---|---|---|---|
| $R^2$ | 0.9200 (N=8) | 0.3017 (N=8) | 0.2577 (N=8) | 0.4061 (N=8) | – (N=5) | – (N=3) | 0.3566 (N=8) |
| RMSE (km) | 2005 (N=8) | 7750 (N=8) | 5797 (N=8) | 4710 (N=8) | 6412 (N=5) | 9614 (N=3) | 5460 (N=8) |
| SI | 15 % (N=8) | 59 % (N=8) | 44 % (N=8) | 36 % (N=8) | 46 % (N=5) | 95 % (N=3) | 41 % (N=8) |
| MAPB | 18 % (N=8) | 80 % (N=8) | 58 % (N=8) | 44 % (N=8) | 70 % (N=5) | 132 % (N=3) | 51 % (N=8) |
| MAPB$_{fair-weather}$ | 15 % (N=4) | 46 % (N=4) | 25 % (N=4) | 18 % (N=4) | 8 % (N=2) | – (N=0) | 9 % (N=4) |
| MAPB$_{windy}$ | 18 % (N=4) | 58 % (N=4) | 40 % (N=4) | 33 % (N=4) | 43 % (N=3) | 74 % (N=3) | 40 % (N=4) |
| Data source (access in June 2022) | https://gulfhypoxia.net/ (Turner et al., 2006; 2008; 2012) http://scavia.seas.umich.edu/hypoxia-forecasts/ (Scavia et al., 2013) https://www.vims.edu/research/topics/dead_zones/forecasts/gom/index.php (Forrest et al., 2011) https://obenour.wordpress.ncsu.edu/news/ (Obenour et al., 2015) https://memg.ocean.dal.ca/news/ (Laurent and Fennel, 2019), https://www.noaa.gov/news (NOAA ensemble) | | | | | | |

[Figure]

**Figure 9. 2015–2020 mean (except for 2016) of predicted HA in scenarios of different nutrient load reduction strategies given different sets of predictors considered. Predictions by the ensemble model are conducted individually for the Shelf-wide cruise periods in different summers and averaged from 2015 to 2020. Horizontal bars indicate ranges of 95 % PIs. Grey dashed lines represent the goal of 5,000 km² set by the Mississippi River/Gulf of Mexico Hypoxia Task Force. Note here nutrient reduction percentages are referred to mid-June nutrient loads in corresponding years.**

**Specific Comments**

Line 15: It may benefit the reader to include a percentage value in comparison to the low RMSE value of 3204 square kilometers, which may be quite large in other coastal systems.

Response: We added three statistics to illustrate the model performances, i.e., percentage difference, mean absolute percentage bias (MAPB, Eq. (14)), and scatter index (SI, Eq. (15)). They can be found in Table 2 and Table 4.

$$MAPB = \frac{1}{N}\sum_{i=1}^{N}\left|\frac{P_i - O_i}{O_i}\right| \times 100\% \qquad (14)$$

$$SI = \frac{RMSE}{\bar{O}} \times 100\% \qquad (15)$$

Line 20: Suggest removing the words "by far". Because this model is the first to do this, the modifier "by far" suggests that no other groups are anywhere near this operational capability. I'm not sure if this is the intent, maybe this is meant instead to say that this ensemble model has the highest performance skill "by far".
Response: We have removed the phrase "by far".

Line 25: Suggest changing to "shelf-wide" here and elsewhere in the paper
Response: We have corrected it accordingly.

Line 30: I've seen "destruction" of hypoxia used more often than "deconstruction" in the literature, suggest making this change
Response: We have corrected it accordingly.

Line 41-43: Awkward phrasing, cut out "however" from sentence
Response: The word "however" has been removed.

Line 46-47: Suggest rephrasing as "An additional Bayesian model applied to summer bottom DO predictions accounts for May total nitrogen…"
Response: We have rephrased it accordingly.

Line 49-52: Suggest rewording as "Mechanistic prediction methods have also been applied by Laurent and Fennel (2019) to develop a weighted mean forecast that is calibrated using May nitrate loads and three-dimensional hindcast simulations over the period 1985-2018. Once calibrated, the model only requires May nitrate loads as an input to produce the seasonal forecast for a given year."
Response: We have reworded these sentences accordingly.

Line 55: Suggest changing "shortages" to "drawbacks"
Response: We have rephrased it accordingly.

Line 55-59: Remove periods before points 2 and 3, otherwise you can remove the colon and break them all up into single sentences. Point 2 could also be reworded slightly, reads awkwardly now. Change "year-to-year" to "interannual"
Response: The periods has been removed. Points are not stated in individually sentences. They are excerpted as followed.
   "(1) The effects of water column stratification are considered only implicitly by the associated wind speeds, water transport, and riverine nutrient loads (usually correlated to river discharges), although stratification is documented as a crucial factor in regulating HA variability. (2) Forecast of the predictors is usually limited, which restricts some of these seasonal models to pseudo ones. (3) Most models are only capable of capturing interannual HA variability and are not reliable in summers when winds are strong."

Line 61-62: Suggest rewording to something like "Here we aimed to provide a new technique in HA prediction that considers both stratification and biochemical effects, and accurately produces daily forecasts of HA based on selected predictors' own forecasts."
Response: We have reworded this sentence accordingly as followed.

"In this study we aim to provide a novel HA prediction method that considers both stratification and biochemical effects. Our new model aims to produce daily HA forecasts based on selected predictors' forecasts with a minimum computational cost."

Line 65-67: Hypoxic volume really hasn't been mentioned up to this point in the manuscript, and here you say that it will be neglected because HA is a better predictor anyway. Would suggest removing these sentences altogether.
Response: We have removed these sentences accordingly.

Line 71-77: I understand that some of the data used for model evaluation are described in the companion paper, but this section seems to be much more focused on derived model inputs (e.g. reanalyses and model outputs). Suggest changing the title of this section to reflect this better.
Response: We have changed the title as "Data preparation".

Line 87: Suggest changing to "… the amount of energy per volume required to homogenize the entire water column"
Response: We have rephrased this sentence accordingly.

Line 95: Change "… are other two factors influencing" to "are two other factors that influence"
Response: We have changed it accordingly.

Line 95-96: Could be worth mentioning that the effect of tidal mixing on stratification is neglected in this study site, since it's included as an additional term in the Simpson 1981 paper.
Response: Yes indeed. The Simpson 1981 paper did consider a tidal mixing term which is ignored in our study. We added a sentence here as followed.

"The tidal effects considered in Simpson (1981) are neglected here due to the relatively weaker contribution in stratification in the shelf when compared to the effects of rivers and winds."

Line 98: The first term on the right-hand side of this equation is negative in Simpson et al. (1978), but it seems like the way that this has been defined (reversing the position of water density and depth-integrated water density), that this may actually be referencing the equation of Simpson 1981. Equation 1 in Simpson 1981 also does not have "h" in the first right-hand side term, but I'm unsure if this is an error on Simpson's part since it appears in the 1978 paper. Suggest changing the reference and/or modifying the equation (may be easier just to change the reference rather than redo calculations/figures).

Response: The "h" term in the first right-hand-side term should not be there if following Simpson's (1981) work. The potential energy anomaly in Simpson (1981) is a depth-averaged term while that in the Simpson et al. (1978) paper is not. We follow the potential energy anomaly equation in Simpson (1981). We have removed the "h" in our equation and rebuilt our model and replotted all figures according to the new results. The Eq. (2)–(3) has been updated

as followed.

$$\frac{d(PEA)}{dt} = \frac{\alpha g}{2c} Q - \delta k_a \rho_a W^3, \tag{2}$$

$$d(PEA)_{heat} \propto Q, \tag{3}$$

Line 110-111: Suggest referencing figure 1a here as was done in lines 90-92.
Response: We have added the referencing figure 1a here.

Line 126: Suggest changing "… estimated for the following" to "estimated by"
Response: We have corrected this sentence accordingly.

Line 128: I am having trouble understanding why this equation does not match what is shown in equation 2.27 of Monteith and Unsworth (2014). It looks as if some simplification occurred such that the denominator of the exponential (T-T', where T'=36K in Monteith and Unsworth) was incorporated into the numerator in the manuscript. However, when I plot the two curves against each other I find that they are unequal, and the gap increases with increasing temperatures. At 20 degrees C, for example, this is equal to vapor pressure difference of approximately 23 Pa. Is this a relatively minor difference, or is this likely to strongly affect the correlation found when combined with W^3?
Response: Thanks for pointing it out. We did miss a T' in the numerator of the exponential term. We have corrected the related equations as followed.

$$p_{sat} = 611 e^{\frac{17.27(T-237.3)}{T-T'}}, \tag{6}$$

where $T' = 36$ K.

$$\rho_a = \rho_a(T,p) = \frac{pM_d}{RT}\left[1 - \frac{611}{p}\left(1 - \frac{M_v}{M_d}\right) e^{\frac{17.27(T-237.3)}{T-T'}}\right], \tag{7}$$

The W$^3$ and the corrected $\rho_a$ are still found significant correlated. The corrected figure shown below has been updated in the manuscript.

[Figure]

**Figure A2. A scatter plot of $\rho_a W^3$ against $W^3$ and their linear correlation.**

Line 142-143: Here I would also suggest pointing the reader to figure 1a as was done in lines 90-92.

Response: Figure reference has been added here.

Line 145-146: Suggest changing phrasing to "However, global forecast model systems like HYCOM do not currently include biochemical fields."

Response: We have rewritten this sentence accordingly as followed.

"However, global forecast models such as HYCOM do not cover biochemical parameters."

Line 156: Suggest removing this sentence and adding the correlation metric to the sentence that describes it first from lines 153-155. This earlier sentence could then read "… calculated as 19 days (R^2=0.8157, Figure A2a)."

Response: We have adjusted these sentences accordingly.

Line 158: Is there a reference for this decomposition rate coefficient, or has this described in more detail in the companion manuscript?

Responses: The SOC is modeled in the accompanying paper (in Eq. (8) and Eq. (10)) proportional to sedimental organic matter concentration (estimated as sedimental particulate organic nitrogen, $PON_{sed}$, and is output from the 3-D coupled model in the accompany paper) and a temperature-dependent decomposition rate. We have added this equation in the manuscript as Eq. (9).

$$SOC = PON_{sed} \cdot VP2N_0 \cdot e^{K_{P2N} \cdot T_b}$$

where $VP2N_0$ is a constant representing the decomposition rates of $PON_{sed}$ at 0 ºC, $K_{P2N}$ a constant (0.0693 ºC$^{-1}$) indicating temperature coefficients for decomposition of $PON_{sed}$, and

$T_b$ the bottom water temperature. In this study, we use the variation of Mississippi River inorganic nitrogen loads with some leading days to mimic the variation of $PON_{sed}$ and keep the temperature-dependent decomposition rate same as that in the 3-D coupled model. Such decomposition rate follows the Q10 assumption (van't Hoff, 1898) that the reaction rate, R, depends exponentially on temperature, i.e.,

$$R = R_0 \cdot Q_{10}^{(T-T_0)/10}$$

For most biological systems, Q10 is from 2 to 3 (Bryan et al., 2008). Here, we assume it as a constant 2. $R_0$ is the reaction rate at temperature $T_0$ (measured in ºC). The SOC scheme we applied takes the $R_0$ as $VP2N_0$ and $T_0$ as 0 ºC. Thus, the above equation can be simplified as:

$$R = R_0 \cdot 2^{\frac{T_b}{10}} \approx R_0 \cdot e^{0.0693 \cdot T_b}$$

van't Hoff, J. H. (1898). Lectures in theoretical and physical chemistry: Part I: Chemical dynamics LK (p.256). London: Edward Arnold. https://UOLibraries.on.worldcat.org/oclc/220605730

Reyes, B. A., Pendergast, J. S., & Yamazaki, S. (2008). Mammalian peripheral circadian oscillators are temperature compensated. Journal of biological rhythms, 23(1), 95–98. https://doi.org/10.1177/0748730407311855

Therefore, we rewrote this part as followed.

The exponential term in Eqs. (9)–(10) estimates the temperature-dependent decomposition rate of organic matter.

$$SOC = PON_{sed} \cdot VP2N_0 \cdot e^{K_{P2N} \cdot T_b}, \tag{9}$$

$$SOCalt = \text{Mississippi River inorganic nitrogen loads (led by 44 days)} \cdot e^{0.0693 T_b}, \tag{10}$$

$VP2N_0$ is a constant representing the decomposition rates of sedimentary particulate organic nitrogen, $PON_{sed}$, at 0 ºC. $K_{P2N}$ is a constant (0.0693 ºC$^{-1}$) indicating temperature coefficients for decomposition of $PON_{sed}$. $T_b$ is bottom water temperature (in ˚C). The Q10 (= 2 given the above chosen coefficients) assumption is applied to mimic the aerobic decomposition rate of $PON_{sed}$.

Line 163-165: I would suggest immediately describing these variables as PEA_heat, PEA_wind, and DCP_temp, rather than defining them here again.
Response: These variables have been described as they are derived in sections 2.1.1 and section 2.1.2. We rewrote these two sentences to avoid prolixity.
    "As listed in Table 1, six candidate predictors are considered in the statistical models including four stratification-related variables (PEA, SSS, PEA$_{heat}$, and PEA$_{wind}$) and two bottom biochemical variables (SOCalt and DCP$_{Temp}$)."

Line 166: Can you better define what it means when you state that "multicollinearity may become a problem"? Maybe adding a short technical detail on the ramifications of this would be helpful to the reader.
Response: The multicollinearity indicates correlation among independent variables. When the multicollinearity problem occurs (i.e., strong correlations among independent variables are

found), the assumption of independent variables is weakened or even collapses. It would lead to unreasonable coefficients for some highly correlated "independent" variables even though the response is well fitted by regression models. Thus, we should avoid this problem or make it less apparent. We did a best-subset searching for predictors and finally found three predictors (PEA, SOCalt, and $DCP_{Temp}$) that provided the best performance when applied to the GLMs and GAMs. The variance inflation factors (VIFs) of these predictors are as 2.15, 2.70, and 1.59, respectively, which is less than 5 suggesting the violation of multicollinearity is negligible.

We thus added a brief description for this issue.

"The multicollinearity can harm the assumption that predictors are independent. It can lead to difficulties in individual coefficients test and numerical instability (Siegel and Wagner, 2022)."

Line 169-170: Are all the grid cells the same size for this model domain? Is this described in more detail in the companion paper?
Response: The sizes of the grid cells are not the same but are nearly a constant of $25.56\pm0.17$ $km^2$ (mean $\pm$ 1std). The minimum and maximum sizes are 25.18 $km^2$ and 25.96 $km^2$, respectively. We thus rewrote this sentence as followed.

"The HA is estimated by the number of hypoxia cells (ROMS computational cells reaching hypoxic conditions) times a nearly constant value (area of the computational cell), which is $25.56 \pm 0.17$ $km^2$ (mean $\pm$ 1SD)."

Line 188: Change "rest" to "remaining"
Response: We have changed it accordingly.

Line 190-191: Change "is chosen randomly" to "are chosen randomly" and "is grouped into" to "are grouped into"
Response: We have corrected them accordingly.

Line 192: Suggest changing to "split at intervals of 5000 km^2"
Response: We have changed it accordingly.

Line 272: Some awkward phrasing "… which impose more threatens to the shelf ecosystem."
Response: We wanted to emphasize that it is more important to increase the model performance in the hypoxic area peak during which the shelf ecosystem would face more threats than during the mild hypoxic events. We have rewritten this sentence.

"Instead of the prediction performance at non-peak HA, here we focused more on the forecasts at HA peaks which impose more threats to the shelf ecosystem."

Line 299: Misspelling of "procedure"
Response: We have corrected it accordingly.

Line 332-333: Suggest change to "… tends to underestimate HA peak estimates (like those seen at samples 310 and 920)"
Response: We have changed it accordingly.

Line 351-352: What daily data are referred to here, the outputs derived from HYCOM or the nitrate and nitrite loadings from USGS?
Response: The daily data here are the HyCOM data and USGS nitrate and nitrite loads. We have rewritten these sentences as followed.

"The Mississippi River total nitrate+nitrite loadings are provided by USGS NWIS as described in section 2.1.2. Daily HYCOM-derived hydrodynamics and USGS river nitrogen loads from 1 January 2007 to 26 August 2020 are used to reconstruct predictors of PEA, SOCalt, and DCP$_{Temp}$."

Line 378-381: These two sentences are a bit repetitive and could be combined. I'm also not entirely clear about whether HYCOM is expected to integrate USGS runoff in the future. Is the use of daily estimates part of long-term plans for HYCOM simulations?
Response: We have rewritten these sentences as followed.

"These results indicate that the ensemble model can produce a highly accurate prediction for HA summer peaks once water stratification is well resolved. Instead of using monthly river forcings, the HYCOM model may possibly resolve the shelf hydrodynamics by utilizing daily river discharges of the Mississippi and the Atchafalaya Rivers."

Line 399: Some awkward phrasing, suggest changing to "… HA forecast capable of explaining up to 80% of the total variability"
Response: We have corrected it accordingly as followed.

"The ensemble model is capable of explaining up to 77 % of the total variability of the hindcast HA and also provides a low RMSE of 3,256 km$^2$ and low MAPBs for overall (29 %) and peak predictions (25 %) when compared to the daily ROMS hindcasts."

Line 404: "… on HYCOM,s"
Response: We have corrected it accordingly.